# Dynamic sex chromosome expression in *Drosophila* male germ cells

Sharvani Mahadevaraju[1,10], Justin M. Fear [1,10], Miriam Akeju[2,10], Brian J. Galletta[3], Mara M. L. S. Pinheiro [4], Camila C. Avelino[4], Diogo C. Cabral-de-Mello[5], Katie Conlon[2], Stafania Dell'Orso [6], Zelalem Demere[2], Kush Mansuria[2], Carolina A. Mendonça[4], Octavio M. Palacios-Gimenez[4,8], Eli Ross[2], Max Savery[1], Kevin Yu[2], Harold E. Smith [7], Vittorio Sartorelli [6], Haiwang Yang[1,9], Nasser M. Rusan[3], Maria D. Vibranovski[4,11], Erika Matunis[2,11] & Brian Oliver [1,11✉]

Given their copy number differences and unique modes of inheritance, the evolved gene content and expression of sex chromosomes is unusual. In many organisms the X and Y chromosomes are inactivated in spermatocytes, possibly as a defense mechanism against insertions into unpaired chromatin. In addition to current sex chromosomes, *Drosophila* has a small gene-poor X-chromosome relic (4th) that re-acquired autosomal status. Here we use single cell RNA-Seq on fly larvae to demonstrate that the single X and pair of 4th chromosomes are specifically inactivated in primary spermatocytes, based on measuring all genes or a set of broadly expressed genes in testis we identified. In contrast, genes on the single Y chromosome become maximally active in primary spermatocytes. Reduced X transcript levels are due to failed activation of RNA-Polymerase-II by phosphorylation of Serine 2 and 5.

[1] Laboratory of Cellular and Developmental Biology, National Institute of Diabetes and Kidney and Digestive Diseases, National Institutes of Health, Bethesda, MD 20892, USA. [2] Department of Cell Biology, Johns Hopkins University School of Medicine, 725 N. Wolfe Street, Baltimore, MD 21205, USA. [3] Cell Biology and Physiology Center, National Heart, Lung and Blood Institute, National Institutes of Health, Bethesda, MD 20892, USA. [4] Department of Genetics and Evolutionary Biology, Institute of Biosciences, University of São Paulo, SP 05508-090 São Paulo, Brazil. [5] Instituto de Biociências/IB, Departamento de Biologia Geral e Aplicada, UNESP—Universidade Estadual Paulista, Rio Claro, São Paulo 13506-900, Brazil. [6] Laboratory of Muscle Stem Cells and Gene Regulation, National Institute of Arthritis and Musculoskeletal and Skin Diseases, National Institutes of Health, Bethesda, MD 20892, USA. [7] Genomics Core, National Institute of Diabetes and Kidney and Digestive Diseases, National Institutes of Health, Bethesda, MD 20892, USA. [8] Present address: Department of Evolutionary Biology and Department of Organismal Biology, Systematic Biology, Evolutionary Biology Centre, Uppsala University, 75236 Uppsala, Sweden. [9] Present address: Department of Pharmacology, Feinberg School of Medicine, Northwestern University, Chicago, IL 60611, USA. [10] These authors contributed equally: Sharvani Mahadevaraju, Justin M. Fear, Miriam Akeju.. [11] These authors jointly supervised this work: Maria D. Vibranovski, Erika Matunis, and Brian Oliver. ✉email: briano@niddk.nih.gov

Organisms as diverse as flies and humans have XY males and XX females. In *Drosophila*, the Y chromosome is large, heterochromatin-rich, and gene poor. It carries genes essential for fertility, but not viability[1,2]. The X chromosome is present as a single copy in males. The small 4th chromosome is an ancestral X chromosome[3,4] that is now disomic. Given the sex chromosome copy number differences and their unique modes of inheritance, the unusual evolved gene content and expression of sex chromosomes is widely studied[5]. In many organisms, including mammals[6], *C. elegans*[7], and *Drosophila*[8], X expression is reduced in testis. Non-mutually exclusive reasons for this reduction include: inactivation of unpaired chromosomes[9,10], absence of upregulation of the X chromosome in the germline (canonical *Drosophila* dosage compensation)[11], and sexually antagonistic selection followed by evolutionary relocalization of genes required in males from the X chromosome to the autosomes[8,12–14]. Some combination of these models is possible, if not likely.

Meiotic X-chromosome inactivation is well described in mammals, where premature transcriptional inactivation of the X and Y chromosome during mid-spermatogenesis is essential for fertility[15]. The X and Y preciously condense into a heterochromatic "XY body" within the pachytene spermatocyte nucleus. Transcriptional inactivation is thought to protect these largely dissimilar chromosomes by defending un-synapsed regions from stealthy invasive transposons that cannot be detected as foreign in the absence of a homolog[16]. Inactivation may also protect against unwanted recombination between the X and Y, or to repair damage created by lack of recombination[17]. Inactivation might also mark paternal X chromosomes for later imprinting for inactivation[18,19]. Some of these models are unlikely in the case of *Drosophila* males, which completely lack meiotic recombination[20]. Exploiting differences, and commonalities, between *Drosophila* and other models can illuminate causes of unusual sex chromosome expression patterns.

In many organisms, the potentially lethal imbalance of X-linked gene transcripts from the single X relative to the paired autosomes must be alleviated in the soma to ensure viability[21]. In *Drosophila* somatic cells, X chromosome dosage compensation is essential and involves large-scale transcriptional upregulation of the single X chromosome[22]. It is unclear if X chromosome dosage compensation exists in *Drosophila* germ cells, which do not require the core dosage compensation machinery used in the soma[23,24]. If dosage compensation occurs, it relies on a undiscovered mechanism. Analysis of transcripts expressed in portions of hand-dissected testes[25,26] or in mutants that block spermatogenesis in mitotic stages[13,21,27] suggest that *Drosophila* testes enriched in spermatogonia show upregulation of the X, consistent with dosage compensation in pre-meiotic cells. Portions of adult testes enriched in late stage spermatocytes[25] show greatly reduced X chromosome expression and autosomal genes inserted on the X show dramatically reduced expression in spermatocytes, far exceeding simple failure of dosage compensation[23,24,28], consistent with X inactivation in primary spermatocytes.

Given the dramatic differences in the gonads and gametes between the sexes, the optimal male and the optimal female genome will differ. For autosomes, which reside in each sex in equal dose, selection is balanced. In stark contrast, sex chromosome residency is not balanced. In a population with equal numbers of males and females, 2/3rds of X chromosomes reside in females. The X chromosome residency profile is expected to result in more opportunities for selection of alleles favoring females. The Y, of course, resides only in males and is under selection only in males. The presence of a homolog is also important. The single X and single Y chromosomes in males are under immediate selection, while only alleles with some degree of

dominance are immediately selected in females. Assuming that there is at least subtle dominance[29], then the X chromosome should be both feminized and demasculinized (alleles with female advantage selected for, and alleles with male advantage selected against), and the Y should be both masculinized and defeminized[8,13]. These patterns have been observed in *Drosophila* species, where expression from the X chromosome is reduced, and where genes required in males have evolutionarily relocated to other chromosomes[12]. Evolutionary arguments for sex chromosome gene content and expression present an interesting causality dilemma. In the model, antagonistic selection for female functions on the X drives removal of genes that males need for development from the X. Removal of those genes is permissive for events such as X inactivation in the male germline[18]. It then follows that X inactivation in the male germline would provide even more selective pressure against X genes with male-biased functions. In this work, we show that sex chromosome expression is dynamic at tissue and single cell resolutions.

## Results

**Sex chromosome expression in different tissues.** *Drosophila* have X and Y sex chromosomes, two major autosome pairs and a pair of "dot" 4th chromosomes (Fig. 1a). The Y and 4th chromosomes are gene poor, while the remaining chromosome arms are gene rich (Fig. 1b). To examine sex-biased gene expression patterns, we focused on the distribution of male-biased gene expression across chromosomes or chromosome arms (chromosome elements) for each tissue. We measured adult gene expression (quadruplicates) in the whole body (Fig. 1c) as well as seven tissues (Fig. 1d–j): head, thorax (viscera removed), abdomen (viscera and all reproductive organs removed), viscera (including digestive and excretory organs), reproductive tract (gonads and genitalia removed), terminalia (including genitalia and analia), and gonads in females and males from two strains. We found a significant deviation from random distributions of sex-biased gene expression (we use $p \leq 0.01$ throughout this study) in whole body, head, thorax, viscera, reproductive tract, and gonad ($\chi^2$ test of independence, Supplementary Dataset 1). Sex chromosomes and former sex chromosomes are the major contributors to the non-randomness.

Overall gene expression of the sex chromosomes varied by tissue. For X-chromosomes, we observed under-expression of genes in the whole body and gonads of males from either of two wildtype strains (Fig. 1c, j), as previously reported[8]. We also observed reduced X-chromosome expression in heads, thorax, reproductive tract, and terminalia (Fig. 1d, e, h). These non-gonadal patterns of X chromosome expression are difficult to explain by absence of germline X chromosome dosage compensation or meiotic sex chromosome inactivation, since there are no germ cells in those tissue. By elimination, this suggests that sexual selection drives gene expression patterns of X-chromosome expression in many somatic cell types. Under-representation in gonads could be explained in full or part by the absence of dosage compensation or meiotic sex chromosome inactivation.

Males with no Y chromosome are viable, but sterile and the Y chromosome is known to be expressed in spermatocytes[30]. However, the tissue-specific Y chromosome gene expression pattern is poorly described. We report that Y-chromosome gene expression was detectable only in whole males and gonads (Fig. 1c, j).

The 4th chromosome showed a decrease in gene expression in male whole bodies, reproductive tracts, and gonads (Fig. 1c, h, j). As a former X chromosome, 4th chromosome expression in the gonads mirrored the X-chromosome underrepresentation of male-biased gene expression. Additionally, and unlike the X chromosome, 4th chromosomes are present in two copies in

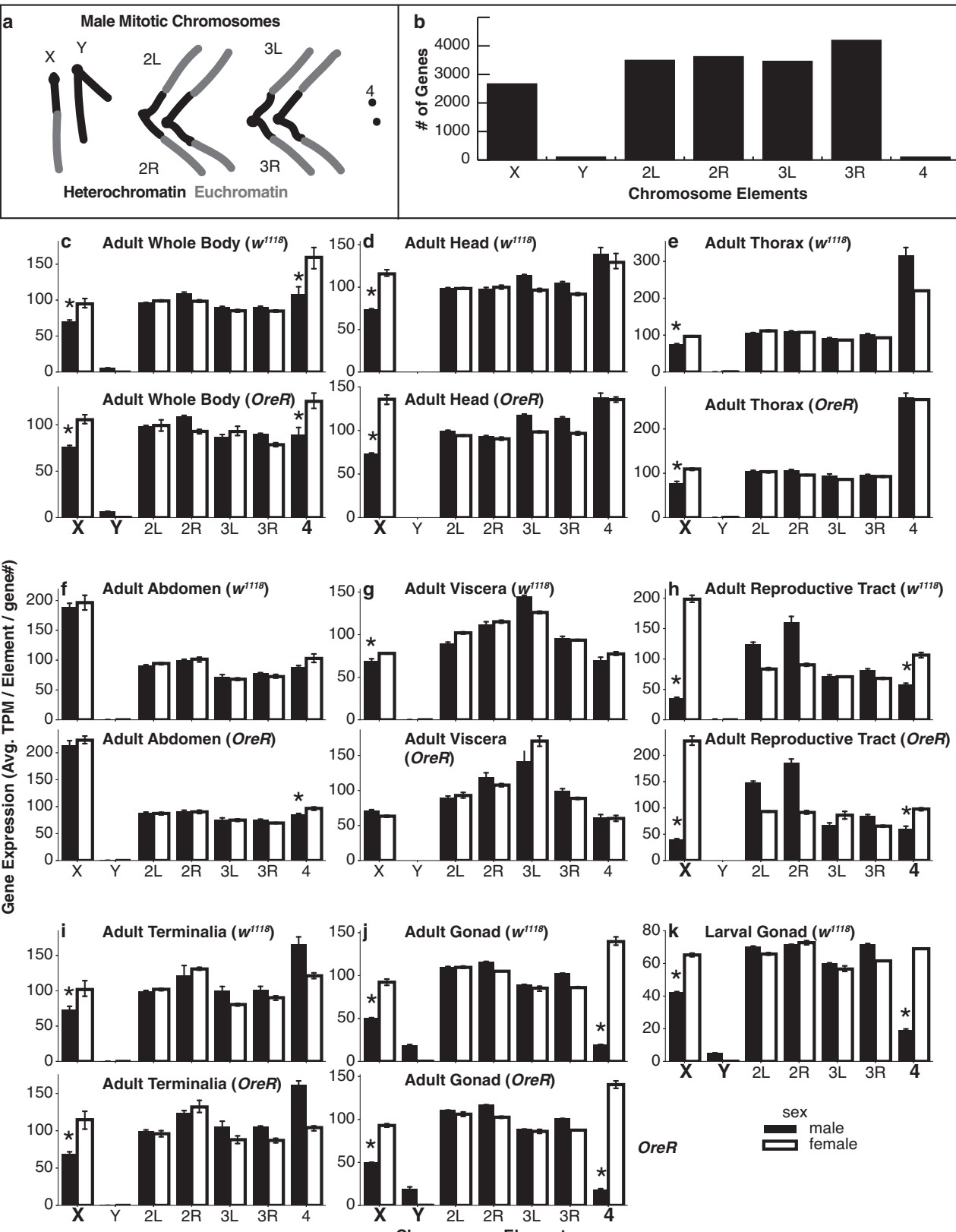

males. Because there are two copies of the 4th chromosome genes, under-representation of expression cannot be explained by the absence of dosage compensation. These data again suggest that there has been sexual selection of sex chromosomes, but only gonads show sex-biased expression of the *Drosophila* X, Y, and 4th chromosomes.

Given the dynamic expression of the sex chromosomes by tissue, it is reasonable to assume that there could be differences by cell type and stage within the gonad. Examination of X-linked gene expression from hand-dissected anterior, middle, and posterior regions of adult testes shows a dynamic pattern of X chromosome expression that decreases as germ cells mature[25].

**Fig. 1 Bulk RNA-Seq of seven adult tissues and L3 larval gonads. a** Illustration of a wild-type male karyotype cartoon depicting the size of the chromosomes and arms (chromosome elements) and the distribution into blocks of heterochromatin (black) and euchromatin (gray). **b** Haploid annotated gene content of chromosome elements (including non-coding genes) based on FlyBase genome r6.26. **c–k** For each tissue type, we summed the transcripts per million reads (TPM) of each gene on a chromosome element and divided by the number of genes expressed on that arm. Male (black bars) and female (open bar) gene expression with bootstrapped 95% confidence intervals of the mean (error bar) are shown. Significant reductions in male relative to female expression (FDR corrected $t$-test for independent samples; female and male at $p \leq 0.01$) are shown (asterisk). Where the chromosomal expression showed a sex difference in both strains, the chromosome element is highlighted (bold and in a larger font). Adult tissues: **c** whole body, **d** head, **e** thorax (viscera removed), **f** abdomen (viscera, reproductive organs removed), **g** viscera (digestive tract and malphigian tubules), **h** reproductive tract (gonads and genitalia removed), **i** terminalia (genitalia and analia), and **j** gonad. For each tissue, we used two "wild-type" strains, $w^{1118}$ and Oregon-R (Ore-R), which are indicated in each panel. **k** Late third instar larval gonads are from $w^{1118}$. Biological samples were prepared in quadruplicate (**c–k**) for each sex.

We decided to use single cell RNA sequencing (scRNA-seq)[31] for a higher resolution picture. We did not want to use read depth to sequence transcriptionally inactive secondary spermatocytes and sperm, so we selected *Drosophila* third instar larval (L3) testis for our experiments. They contain abundant germ cells, including the critical transition from mitotic spermatogonia to meiotic primary spermatocytes[32,33]. Prior to using L3 testes for single cell analysis, we asked whether the patterns of chromosome-specific gene expression known to occur in adult gonads[8] are also present at this earlier stage of gonadogenesis. We performed RNA-Seq on eight biological replicates each of cleaned 3rd instar larval testes and ovaries (Supplementary Fig. 1) to compare with adult tissues. The overall profiles, including X, Y and 4th chromosome expression are similar in both adult and larval gonads (Fig. 1j, k).

**Single cell resolution gene expression in the testis**. Determining which cells of the testis contribute to unusual patterns of sex chromosome expression is challenging because of the cellular diversity and dynamic nature of gene expression in the gonad. *Drosophila* spermatogenesis (Fig. 2a) begins at the apical end of the testis[34] where germline stem cells divide asymmetrically, generating mitotically active differentiating daughters called gonialblasts that produce clusters of interconnected spermatogonia (G). After their fourth mitotic division, spermatogonia quickly enter pre-meiotic S-phase, becoming early primary spermatocytes (E1°) that are initially morphologically indistinguishable from spermatogonia[35]. As early primary spermatocytes traverse an extended G2 phase, they undergo a burst of transcriptional activity that is accompanied by a 25-fold increase in volume[33], becoming large middle primary spermatocytes (M1°) and then slightly smaller late primary spermatocytes (L1°). Fully mature late primary spermatocytes progress through two rapid meiotic divisions, becoming spermatids. Pairs of quiescent somatic cyst cells (C1–C4) envelope each gonialblast and descendants, differentiating alongside the germ cells they support. Cyst cells are a source of intercellular signals and act as a permeability barrier[36,37]. The entire L3 testis is encapsulated by an epithelial monolayer of pigment cell precursors (P), which become adult pigment cells and are required for joining the gonad with the reproductive tract[38]. The basal end of the L3 testis contains terminal epithelial precursor cells (T) which will ultimately contribute to maturation in the final steps of spermiogenesis[39].

To capture the transcriptional profiles of the cell types in the *Drosophila* L3 testis, we dissected staged male third instar larvae, enzymatically removing the associated fat body before dissociating 20–40 testes to a single cell suspension for scRNA-seq ("Methods", Supplementary Fig. 1a–d). We piloted dissociation conditions using fly strains expressing GFP in subsets of cells (Supplementary Fig. 1g, h) coupled with dye exclusion (Supplementary Fig. 1i) (70–75% viability, $n = 1000$/replicate), ensuring consistent retrieval of single cell suspensions of viable germline and somatic cells. We then performed scRNA-seq on wild-type

L3 testes. We identified 18,965 single cells across three biological replicates based on the intersection of calls from *cell ranger count*[40] and *DropletUtils emptyDrops*[41] (Supplementary Dataset 2). Potential cell doublets were detected using *scrublet*[42] and removed. Based on preliminary cluster analysis using the 2000 most variably expressed genes, we set the perplexity threshold in *Seurat*[43] to 0.3. This yielded ten clusters, each potentially representing a distinct cell type or state with each of three biological replicates contributing to the clusters (Spearman expression rank correlation $\geq 0.93$, $p < 0.01$, Fig. 2b). While the replicates show the absence of large effects, systematic errors in reporting the biology were still possible. If we captured the majority of the cells and cell types, then we should observe a strong correlation between RNA-seq from the whole organ and the total of all single cells. Indeed, the correlation between replicated bulk RNA-Seq from whole L3 testes and sum of single cells from L3 testes was significant (Fig. 2c; Supplementary Dataset 3), indicating that major cell types are well represented in our scRNA-seq dataset. Gene Set Enrichment Analysis GSEA on the ten clusters indicated that genes with male-biased expression in whole gonads are also enriched in germ cells relative to somatic cells in the L3 gonad (compare red and blue plots, Fig. 2d). Thus, the germline is the major contributor to the pattern of sex-biased expression of the X, Y, and 4th chromosomes.

To align gene expression patterns with cell types, we first identified genes with preferential expression in particular cell types, then compared the expression profiles of cells in each of the ten groups with that of all remaining cells in an iterative process for each cell cluster (Fig. 2e, Supplementary Dataset 2). To determine what the cell types were, we projected genes with well-known expression patterns, carefully re-curated from published images (Supplementary Dataset 4), onto the cell level expression profiles. We also used transgenic protein-trap reporter genes[44] to identify expression patterns for genes with enriched expression in each cell type (Fig. 2f–k, Supplementary Fig. 3, Supplementary Dataset 4). This allowed us to annotate all ten clusters. 91% of 79 genes from the literature or reporter expression patterns overlapped cell types identified by scRNA-seq. 46% were expressed in the same or developmentally preceding cell types, and an additional 45% were expressed in the same cell lineage. The 7% of genes showing discordant assay-dependent expression patterns were protein-trap reporters showing either weak or widespread expression, both of which hinder assignment to specific cell types (Supplementary Fig. 3, Supplementary Dataset 4). These data indicate that we have identified many of the major cell types in the L3 testis. We did not identify previously unidentified cell types and failed to unambiguously identify cells in the stem cell niche.

Spermatogonia (G) were characterized by the biased expression of 624 genes (Fig. 2e #1) including: known early germ cell markers *argonaute 3* (*AGO3*), *aubergine* (*aub*), and *vasa* (*vas*)[45], the early differentiation signal *bag-of-marbles* (*bam*)[35], the mitotic cell marker *p53*[46] (Fig. 2f), the chromatin proteins

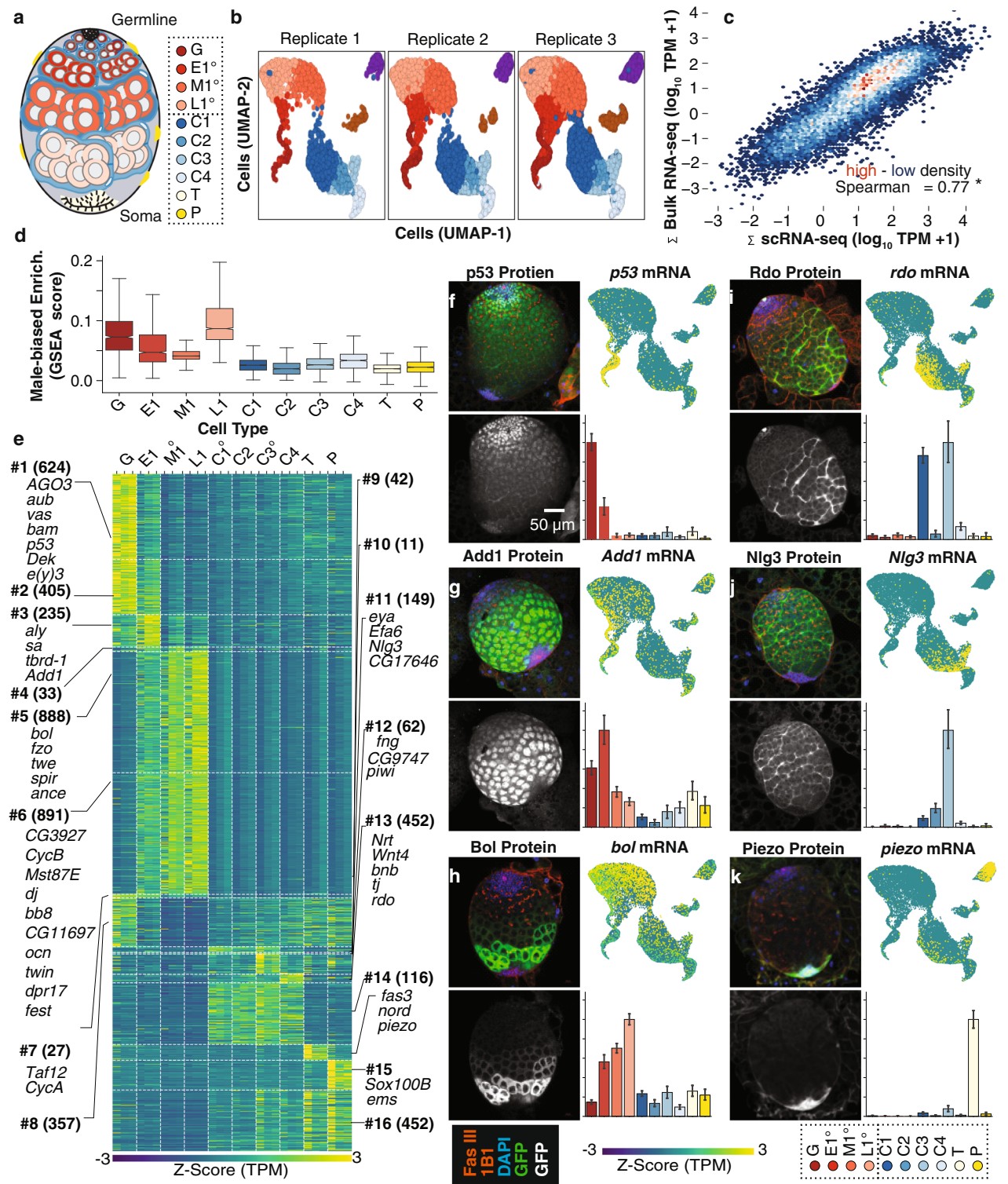

*Dek*[47] and *enhancer of yellow 3* (*e(y)3*)[48] (Fig. 2e #1). While expression of these spermatogonial genes also extended to spermatocytes, we could readily distinguish spermatogonia from spermatocytes by their lack of spermatocyte-specific gene expression. The earliest primary spermatocytes (E1°) were characterized by high expression of 235 genes including three markers of the transition from spermatogonia to spermatocyte: *always early* (*aly*), *spermatocyte arrest* (*sa*)[49], and *testis-specifically expressed bromodomain containing protein-1* (*tbrd-1*)[50] (Fig. 2e #3). Sa is a testis-specific TATA-binding protein associated factor

(tTAF), Aly serves as a testis-specific meiotic arrest complex (tMAC) member[49], and *tbrd-1* is associated with both transcriptional complexes[51]. In addition, we show that *ADD domain-containing protein 1* (*Add1*), which encodes a heterochromatin associated protein that interacts with Heterochromatin Protein 1 (HP1) to maintain heterochromatin[52,53], has enriched expression in early primary spermatocytes in the scRNA-seq data. The Add1 protein accumulated throughout spermatocyte stages, consistent with translational control and/or protein stability over several days of germ cell development (Fig. 2g). The tTAFS and tMACs

**Fig. 2 Identification and annotation of larval testis cell types from scRNA-seq and comparison to bulk testis RNA-Seq. a** Illustration of the 3rd instar larval testis, showing germline cell types spermatogonia (G); early- (E1º), middle- (E2º), and late- (L1º) primary spermatocytes; four groups of somatic cysts cells (C1–C4), terminal epithelium precursors (T) and pigment cells (P). These cell types are from germline or soma linages (dashed boxes). See abbreviation and color code key used throughout this manuscript. **b** Consistency of cell type expression profiles. Uniform Manifold Approximation and Projection (UMAP) for each scRNA-seq replicate. Each cell position in multi-dimensional space projected onto the two-dimensional UMAP space and clustered using k-nearest neighbors. **c** Correlation between whole testis and single cell profiles. Density scatterplot (high density, red and low density: blue); regression line and Spearman rank correlation, $\rho$) of gene expression from the sum of all cells from the larval testis scRNA-seq (x-axis) versus bulk larval testis RNA-Seq (y-axis). Transcripts Per Million reads +1 (TPM + 1). **d** Distribution of genes showing testis-biased expression in bulk RNA-Seq using Gene Set Enrichment Scores (y-axis) among cell types (x-axis) defined by L3 testis scRNA-seq. Each bar is a cell type, so sample size is the number of cells in that cell type: G = 1065, E1 = 1267, M1 = 4570, L1 = 2667, C1 = 4064, C2 = 1230, C3 = 910, C4 = 945, T = 1379, P = 868. Boxplots (box = interquartile range (IQR), notch = 95% 736 confidence interval of median, whiskers = ±1.5xIQR). **e** Patterns of differential gene expression among putative cell types. Only differentially expressed genes are shown (two-sided Wilcoxon rank-sum test, FDR corrected $P \leq 0.01$). Gene expression is summarized (Z-score; low: blue, high: yellow) for each differentially expressed gene (rows) for all cell-type-by-replicate (columns). Genes are ordered and grouped (horizontal dashed lines) manually into 16 classes (numbered) based on differential gene expression. Each class is annotated with the number of genes (parentheses) and the key genes used to identify the cell types (Supplementary Dataset 2). **f–k** Immunofluorescence images showing protein expression of representative genes from the protein trap reporters used for cell type annotation. **f** p53, **g** Add1 **h** bol **i** rdo **j** Nlg3, and **k** Piezo. Each panel (counter clockwise from top left) consists of color (Fas III, 1B1: red, DAPI: blue, and GFP: green) and grayscale (GFP) images of protein trap expression, normalized gene expression values (normalized to maximum expression) plotted by cell type (Each bar is a cell type, so sample size is the number of cells in that cell type: G = 1065, E1 = 1267, M1 = 4570, L1 = 2667, C1 = 4064, C2 = 1230, C3 = 910, C4 = 945, T = 1379, P = 868. Error bars are bootstrapped 95% confidence intervals around the mean), and scRNA-seq gene expression projected onto the UMAP (Z-score; low: blue, high: yellow, see key).

initiate expression of a large battery of genes, many expressed exclusively in primary spermatocytes[54]. As expected, middle and late primary spermatocytes (M1º and L1º) are characterized by the increased expression of two large groups of 1779 genes that are targets of tTAFs and tMACs (Fig. 2e #5, #6), including *don juan (dj)*[55]. One of these genes was *ocnus (ocn). ocn* transgenes show very poor expression when inserted on the X chromosome versus when inserted on autosomes, providing some of the strongest evidence that X inactivation occurs in *Drosophila*[28] (Fig. 2e #6). As spermatocytes progress through differentiation, we observed a concomitant increase in the expression of the meiotic cell cycle regulator *twine (twe)* and the translational regulator of Twe encoded by *boule (bol)*[56,57] by both scRNA-seq and protein-trap analysis (Fig. 2h). These data defined four distinct stages in male germ cell development in larval testis.

The somatic cell types provide an important baseline for analysis of germline expression and contribute to further understanding of these cell types in the testis (Fig. 2e). For example, *traffic jam (tj)* and *eyes absent (eya)* are expressed in early and late cyst cells respectively in both adult and L3 testis[58,59], and are most highly expressed in distinct cyst cell clusters in this study. There is also a great deal of biology in the cell expression profiles. For example, the transmembrane protein encoded by *defective proboscis extension response 17 (dpr17)*, a neuronal surface label, is enriched in spermatocytes. Genes encoding the visual system Reduced ocelli (Rdo) protein[60] (Fig. 2i), the exchange factor for the GTPase Arf-6 (Efa6)[61], and the synaptic adhesion molecule Neuroligin 3 (Nlg3)[62] (Fig. 2j), show dynamic expression patterns in the cyst cells. The terminal epithelium is poorly studied and we report here multiple RNA and protein-trap markers for this cell type, including the homolog of human Neuron Derived Neurotrophic Factor (Nord), as well as the mechanosensory ion channel subunit Piezo[63] (Fig. 2k), which acts as a stretch sensor and could help regulate transit of sperm from the testis into the seminal vesicle. Our data (See Supplementary Fig. 2 for UMAP projections for each *Drosophila* gene), along with a similar dataset from adult testis[64], should be a resource for those studying testis development and physiology.

**Dynamic and cell type specific sex chromosome expression.** We looked at the dynamics of sex chromosome gene expression in germ cells in addition to all the other cell types from the single

cell dataset (Fig. 3, Supplementary Dataset 5). The somatic cell X chromosome expression relative to autosomes hovered near 1.0 despite the 2-fold dose difference, a pattern consistent with the known canonical X chromosome dosage compensation mechanism in somatic cells, expected to increase expression from the single X[22]. This dosage compensation mechanism does not exist in germ cells. Nevertheless, spermatogonia showed similar levels of X chromosome expression relative to autosomes, providing evidence for non-canonical dosage compensation of the X chromosome in pre-meiotic germ cells. There was a significant decrease in expression of the X chromosome in early, middle, and late primary spermatocytes (M1º and L1º) relative to either spermatogonia or somatic cells. This decrease in X expression approached 2-fold which could be due to either a loss of dosage compensation in germ cells as they mature into primary spermatocytes, or to the gain of meiotic X-chromosome inactivation during the transition from mitotic spermatogonia to meiotic spermatocytes. At least some genes on mammalian X chromosomes are also overexpressed prior to inactivation[65], raising the possibility that this dynamic transition in X chromosome expression in the male germline is conserved.

Expression of 4th chromosome genes parallels what was seen for the X (Fig. 3b). The expression ratio of the two 4th chromosomes relative to the two sets of major autosomes hovered near 1. There was a significant decrease in relative 4th chromosome expression in middle and late primary spermatocytes (M1º, and L1º) compared to expression in either spermatogonia or somatic cells. Since we can rule out failed dosage compensation as a cause of 4th chromosome decreased expression, there must be a gain of inactivation during the developmental transition from mitotic spermatogonia to meiotic spermatocytes. This X chromosome like behavior may reflect the evolutionary history of the 4th chromosome; specifically, that the 4th retained X-inactivation after reacquiring autosomal status.

Finally, the entire pattern of Y chromosome gene expression is inverted relative to that seen for the X and 4th chromosomes (Fig. 3c). We observed poor expression of the Y in somatic cells and spermatogonia, and increased expression in primary spermatocytes (E1º, M1º, and L1º), concomitant with decreased expression of the X and 4th chromosomes. This occurs from expression of a 42 transcriptionally active Y-linked genes (Supplementary Dataset 2), consistent with the diffuse chromatin and Y-loops originally identified by cytology of primary spermatocytes[66,67]. The decrease in X and 4th chromosome

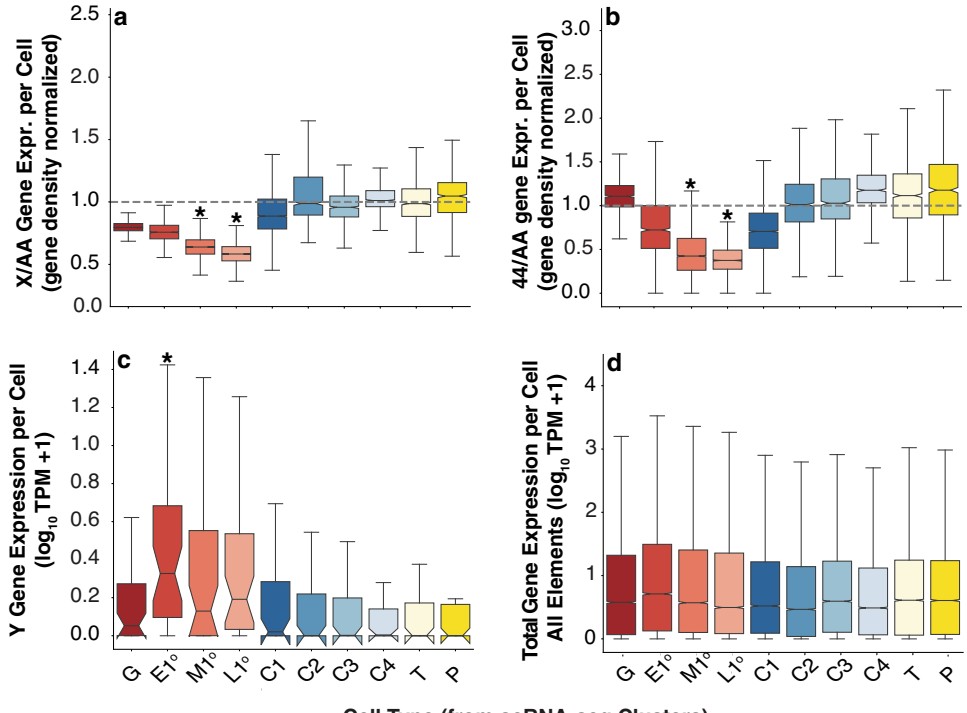

**Fig. 3 X-, 4th-, and Y-linked gene expression patterns across cell types.** Cell type-specific gene expression of **a** X-linked genes and **b** 4th-linked genes scaled by major autosomal element gene expression (2nd and 3rd chromosomes) at cellular resolution [e.g., (X TPM/ #X Genes)/(A TPM/ # A Genes). **c** Y-linked gene expression and **d** genome-wide expression in normalized read counts. Chromosomal distributions in germ cells, significantly different from somatic cell types, are indicated with an asterisk (**a**, **b** $p \leq 0.01$ using permutation test, **c**: $q$-value $\leq 0.01$ using $\chi^2$ test). **a–d** The number of cells scored are: G = 1065, E1 = 1267, M1 = 4570, L1 = 2667, C1 = 4064, C2 = 1230, C3 = 910, C4 = 945, T = 1379, P = 868. Boxplots (box = interquartile range (IQR), notch = 95% confidence interval of median, whiskers = ± 1.5xIQR).

expression in middle and late primary spermatocytes does not reflect an overall decrease in total gene expression compared to somatic lineages, as genome-wide expression was robust in all cell types (Fig. 3d). Since the single X and the two 4th chromosomes showed reduced expression, while the single Y showed ongoing expression, these data demonstrate that there is no simple rule for dose-dependent sex chromosome gene expression in *Drosophila* primary spermatocytes. Genes on chromosomes with no homolog can go up (Y) or down (X), and the 4th appears to retain X-like reduced expression despite having a homolog. These differences in spermatocyte expression fully account for the differences in X, Y and 4th chromosome expression observed in bulk RNA-seq of larval testis.

Since genes with high expression in the testis are not uniformly distributed in the genome[8,13], it was possible that the reduced expression of the X and 4th chromosomes was due to the absence of genes highly expressed in spermatocytes rather than a chromosome-wide reduction in expression due to a more global inactivation. A way to avoid this potential confounding effect, is to explore the expression of widely expressed "housekeeping" genes. We explored three data-driven methods to determine X and 4th chromosome expression of genes with housekeeping functions. In the first two methods, we used low tissue-specificity genes based on Tαυ and Tissue Specificity Score (TSPS) using our data[68,69]. The third method was a more granular low cell-type specificity metric within in the scRNA-seq experiments (CTSP). Specifically, a set of widely expressed genes expressed in ≥33% of all cells. These methods reduced the expressed gene set numbers to varying degrees, with CTSP being the most stringent (Table 1). The Y chromosome was expressed in an exquisitely tissue-specific matter and has no widely expressed genes using any metric. To

**Table 1 Chromosome element genes and expression.**

| Chromosome element | Annotated[a] | Expressed[b] | CTSP[c] | Tαυ[d] | TSPS[e] |
|---|---|---|---|---|---|
| X | 2675 | 2207 | 65 | 336 | 1130 |
| Y | 113 | 42 | 0 | 0 | 0 |
| 2L | 3501 | 2767 | 126 | 367 | 1252 |
| 2R | 3628 | 2861 | 138 | 406 | 1413 |
| 3L | 3466 | 2,867 | 101 | 371 | 1347 |
| 3R | 4201 | 3,467 | 145 | 498 | 1673 |
| 4 | 111 | 102 | 3 | 7 | 54 |

[a]Genes annotated in FlyBase r6.26, including non-coding genes.
[b]Genes detectably expressed in L3 scRNA-seq experiments.
[c]Genes expressed in ≥ 33% of cells in L3 scRNA-seq experiments.
[d]Genes expressed across multiple adult tissues (tissue specificity score $\tau \leq 0.5$).
[e]Genes expressed across multiple adult tissues (tissue specificity score TSPS ≤ 1).

determine if the functions of these three reduced gene sets are consistent with generic gene function, we systematically analyzed Gene ontology (GO) enrichment for all three subsets of genes (Fig. 4a; Supplementary Dataset 5). There are differences in function in the three gene sets. For example, in the Molecular ontology, enzymes were enriched in Tαυ and CTSP gene sets, while regulators (which are less likely to be generic) were more enriched in the Tαυ gene set. In the Biological ontology, all three sets were enriched for protein metabolism, consistent with "housekeeping", but the tissue-level Tαυ and TSPS gene sets were enriched for genes with development and female gamete functions, which is not commonly thought to be generic. Housekeeping genes are often highly expressed. All the reduced gene sets had higher median expression than all expressed genes,

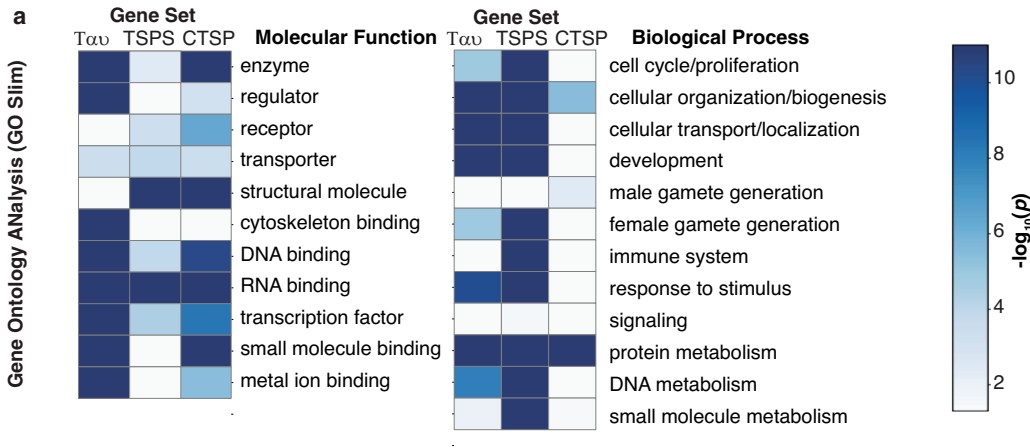

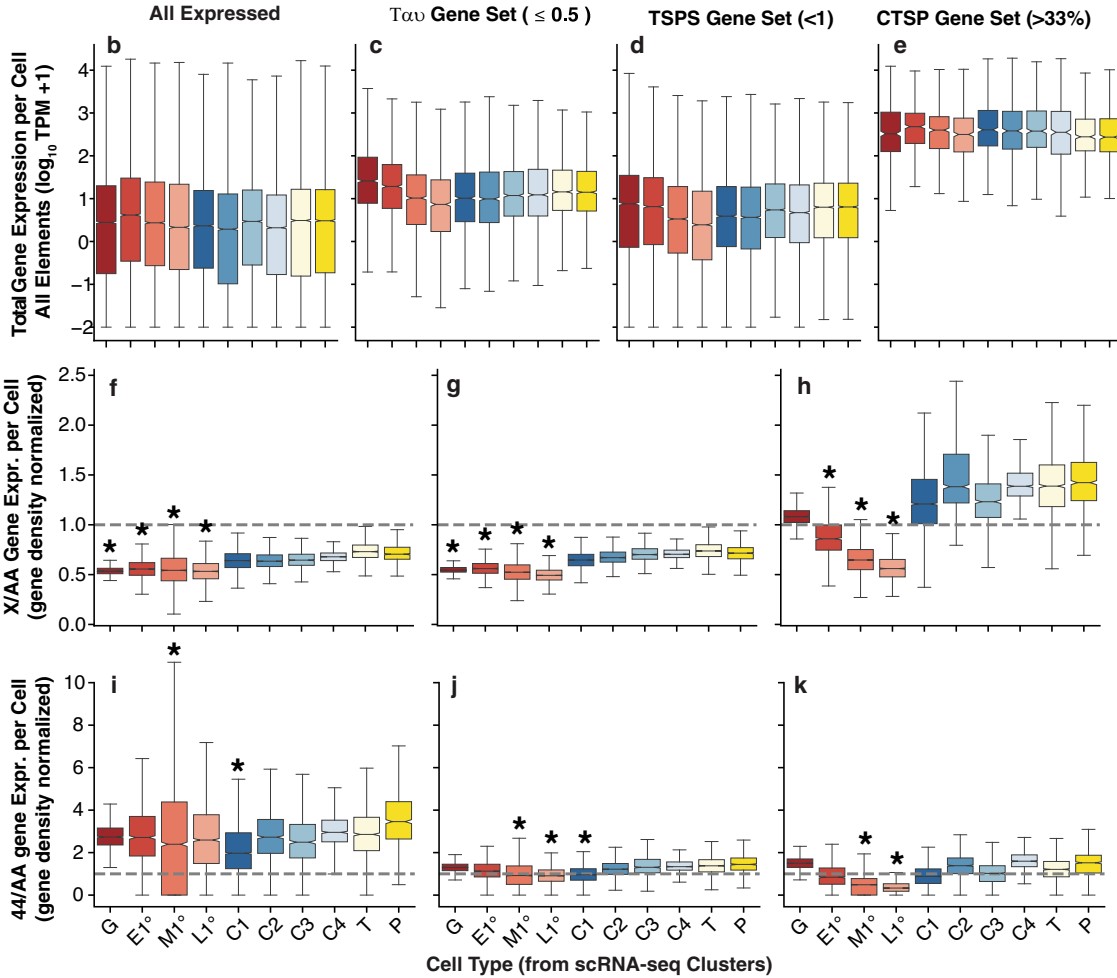

**Fig. 4 Gene expression patterns across cell types and chromosome elements for different gene subsets. a** Ribbons showing p-values for Gene Ontology enrichments (blue scale) in three gene sets representing widely expressed genes. **b–e** Total gene expression (y-axis) by cell-type (x-axis) in L3 testis scRNA-seq for different gene subsets. **f–h** X chromosome, and **i–k** 4th chromosome expression relative to major autosomes by cell type as in Fig. 3a, b. **c**, **f**, **i** Tau: genes not tissue-specifically expressed ($\tau \leq 0.5$ from adult tissues Fig. 1). **d**, **g**, **j** TSPS: genes not tissue-specifically expressed (TSPS ≤ 1 from adult tissues Fig. 1). **e**, **k** CTSP: genes not cell type-specially expressed (≥33% of cells). The number of cells scored are: G = 1065, E1 = 1267, M1 = 4570, L1 = 2667, C1 = 4064, C2 = 1230, C3 = 910, C4 = 945, T = 1379, P = 868. Boxplots (box = interquartile range (IQR), notch = 95% confidence interval of median, whiskers = ±1.5xIQR). Significance at $p \leq 0.01$ determined by permutation testing (asterisk).

but elevated expression was most pronounced in the CTSP gene set (Fig. 4b–e). Additionally, the CTSP gene set showed greater uniformity in expression levels across cell types. Based on these results, we concluded that the CTSP gene set was the best subset for exploring expression of "housekeeping" genes.

We then used the reduced gene sets to examine the expression of the X and 4th chromosomes in all testis cell types. Importantly, when we examined relative expression, all three reduced gene sets showed significantly reduced X/A expression in germline cells (Fig. 4f–h). However, Tau and TSPS gene sets showed reduced X

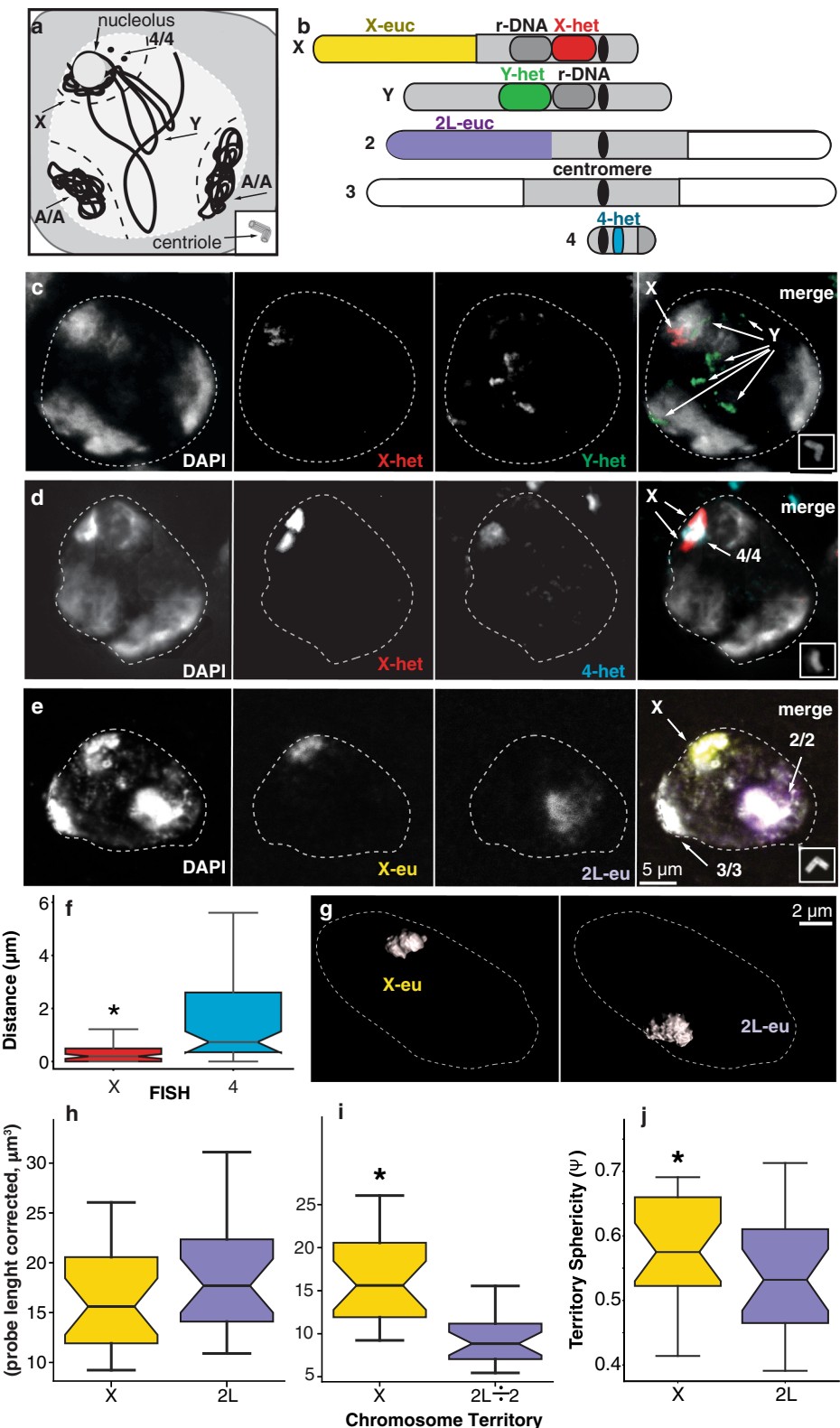

expression in all cell types resulting in X/A ratios approaching 0.5 in both somatic and germline cells (Fig. 4f, g). At face value, failed dosage compensation might be expected to approach 0.5. These observations suggest a reason for the previous conclusion that there is no dosage compensation in male germline[11] following the analysis of widely expressed genes using tissue-specificity scores. This conclusion is likely spurious, as testis somatic cells express

the dosage compensation genes (Supplementary File 1) and the protein complex decorates the X in those somatic cells as occurs in X-chromosome dosage compensation in other somatic cells[23]. In contrast, the CTSP gene set showed reduced X chromosome expression, approaching 0.5, only in the late primary spermato-cytes (Fig. 4h). Spermatogonia and somatic cells showed X/A rations approaching 1.0. Like the analysis of all expressed genes

**Fig. 5 Spatial location of chromosomal territories in late primary spermatocyte nuclei. a** Schematic model summarizing chromosomal spatial location in the spermatocyte nucleus and probe localization on chromosomes. There are two autosomal territories (A/A), X and 4/4 territory close to the nucleolus and Y chromosome. Experimental 1.688 satellite probe to detect X heterochromatin (red), AATAC satellite to detect Y heterochromatin (green), AATAT to detect 4th heterochromatin (cyan), oligo paints to detect X-euchromatin (yellow) and 2 L (purple) are represented on the fly chromosomes (not to scale). **b** DAPI (gray) distinguishing three chromosomal territories where X and Y satellite probes clearly detects the chromosomes, respectively. Primary spermatocyte nucleus is outlined (white-dashed line) based on DAPI. Asterless was used to stage individual cells by progression of centriole elongation (insets). **c**, **d** Spatial location of X and 4th chromosomes. **e**, **g** Representative samples showing spatial location of X and 2 L by oligopaints. **f** Box plots showing the distributions of mean distances between the X (red), or 4th chromosomes (cyan), to the nucleolus ($N = 75$ chromosome pairs). To measure chromosome distance to the nucleolus, we averaged the distance from the outer heterochromatin probe edges to the closest nucleolus point ($p$-value $\leq$ 0.01 two-sided Wilcoxon signed-rank test with continuity correction (asterisk). Images of X and 2 L spatial localization by oligopaints "masked" in Imaris to obtain 3D images to measure the probe-length corrected **h** and the probe-length and copy number corrected **i** volume and **j** sphericity ($N = 23$ nuclei). The sphericity measure accounts for differences in volume: $\phi = ((\pi^(1/3))(6\text{Volume})^(2/3))/(\text{Area})$. **h–j** Sample size was 23. Boxplots (box = interquartile range (IQR), notch = 95% confidence interval of median, whiskers = ± 1.5xIQR). Significance ($p \leq 0.01$, two-sided Welch's $t$-test) (asterisk).

(Fig. 3), the parsimonious explanation is that spermatogonia show dosage compensation and spermatocytes show inactivation or reduced X chromosome compensation.

We similarly examined the expression of the reduced gene sets for the 4th chromosome. Genes with low Tαυ were over-represented on the 4th chromosome, especially in M1º germ cells and C1 somatic cells, resulting in an exaggerated over-expression relative to the major autosomes across all cell types (Fig. 4h), while low TSPS and CTSP resulted in significantly lower relative expression of the 4th chromosomes only in spermatocytes (Fig. 4i, j). The magnitude of spermatocyte decrease was magnified when we used the CTSP gene set, but overall the 4/A ratios were near 1.0 (Fig. 4j). The large sample size of cells resulted in tightly centered distributions, but note that the number of genes contributing the 4th chromosome measurements was small (Table 1). To briefly summarize, we observed a decrease in X and 4th chromosome expression with all genes (Fig. 3) and with reduced gene sets (Fig. 4), suggesting a chromosome-wide change in gene expression in spermatocytes, and not simply a reduced number of X-linked and 4-linked genes with male-biased expression.

**X chromosome condensation in primary spermatocytes.** Given that the X expression falls in maturing primary spermatocytes, we asked if the expression shift in primary spermatocytes correlated with precocious chromosome condensation, as has been previously reported[70]. To locate chromosomes, we used satellite and oligopaint probes that were validated on somatic metaphase chromosomes (Supplementary Fig. 4). The major autosomal bivalents and the X chromosome (along with the 4th chromosomes) reside in three distinct chromosome territories that abut the nuclear envelope of primary spermatocytes[71] (Fig. 5a). The nuclear interior, more diffusely stained by DAPI, was occupied by the large transcriptionally active Y chromosome[66,67,72] (Fig. 5a). The Y chromosome expressed only 42 genes in our experiments, but because of their megabase introns[73], this represents extensive transcription along the length of the chromosome. In situ hybridization reveals X chromosome heterochromatic satellite sequences near the prominent spermatocyte nucleolus, where Ribosomal DNA repeats are located, and ribosome biogenesis occurs (Fig. 5b, c, f)[74]. The only homology between the X and Y chromosome is the rDNA clusters. Only the Y-linked rDNAs are expressed; the X-linked cluster is inactive[75], which is consistent with the pattern we observed chromosome-wide. The probes recognizing satellite sequences of the Y chromosome showed a patchy distribution in the region separating the three main chromosome territories (Fig. 5b, c)[66], consistent with decondensation and transcriptional activity. The 4th is often, but not always, near the nucleolus[72]. We observed that the X was universally near the nucleolus (median distance 0.2 µm) and the 4th

was nearly as close (median distance 0.7 µm), well within the same prominent territory (Fig. 5d, f). Since the 4th occupies the same territory as the X, these chromosomes could be regulated independently, or coordinately, due to territory-level regulation.

The locations of the satellite sequences identify the territories, but we were most interested in chromosome structure in gene rich euchromatic regions. Hence, we used oligopainting to examine the euchromatic portions of the X chromosome for evidence of compaction that might accompany inactivation. Oligopaint probes show that inactive X chromosomes have greater compaction (decreased volume) and increased sphericity compared to active X chromosomes in mammalian cells[72,76], so we measured both these parameters. We probed similar sized euchromatic regions of the X chromosome (22.3 Mb) and the left arm of the 2nd chromosome (2 L, 22.7 Mb) with oligopaints (Fig. 5e). We converted raw in situ data in *Imaris* to create masks ($n = 23$) of pixel intensities (Supplementary Movie 1) and obtained volumetric measurements of the X and 2 territories (Fig. 5g). We found a that probe length corrected X chromosome volume was reduced relative to chromosome 2 L (Fig. 5h). This was not significant at the $p \leq 0.01$ level we have used in this work ($p = 0.03$). However, when we corrected the data to account for 2 L copy number (divided by two), the X had significantly greater volume than 2 L (Fig. 5i) which was inconsistent with inactivity resulting from compaction. The assumption that volume scales with copy number is of dubious validity. The X chromosome was significantly more spherical than 2 L (Fig. 5j), which was consistent with the hypothesis that compaction accompanies inactivation for regulation of X expression in *Drosophila* primary spermatocytes. In mammalian fibroblasts, the inactive X has a sphericity ($\psi$) of 0.67, while the active X has $\psi = 0.57$[72,76]. We found that the *Drosophila* spermatocyte X had $\psi = 0.58$, while 2 L had $\psi = 0.53$. Collectively, these results do not provide strong evidence that *Drosophila* spermatocyte X chromosome activity is regulated by overall condensation levels.

**Inactive RNA polymerase II mechanism.** We more directly addressed transcriptional status of chromosome territories in spermatocytes by determining the phosphorylation status of the regulatory C-Terminal Domain (CTD) of RNA polymerase II (Pol-II). The transcription cycle begins when Pol-II binds the promoter. Serine 5 phosphorylation (pSer5-CTD) causes RNA Pol-II to initiate transcription, and subsequent Serine 2 phosphorylation (pSer2-CTD) induces transcriptional elongation (Fig. 6a)[77]. In the whole testis, we observed pSer2-CTD in spermatogonia and in the two autosomal territories in spermatocytes (Fig. 6b). In spermatocytes, we observed clear pSer2-CTD in two autosomal territories, but the X and 4th territory (identified by nucleolar proximity) was poorly stained as previously reported[78] and pSer5-CTD localization was similar to pSer2-CTD pattern

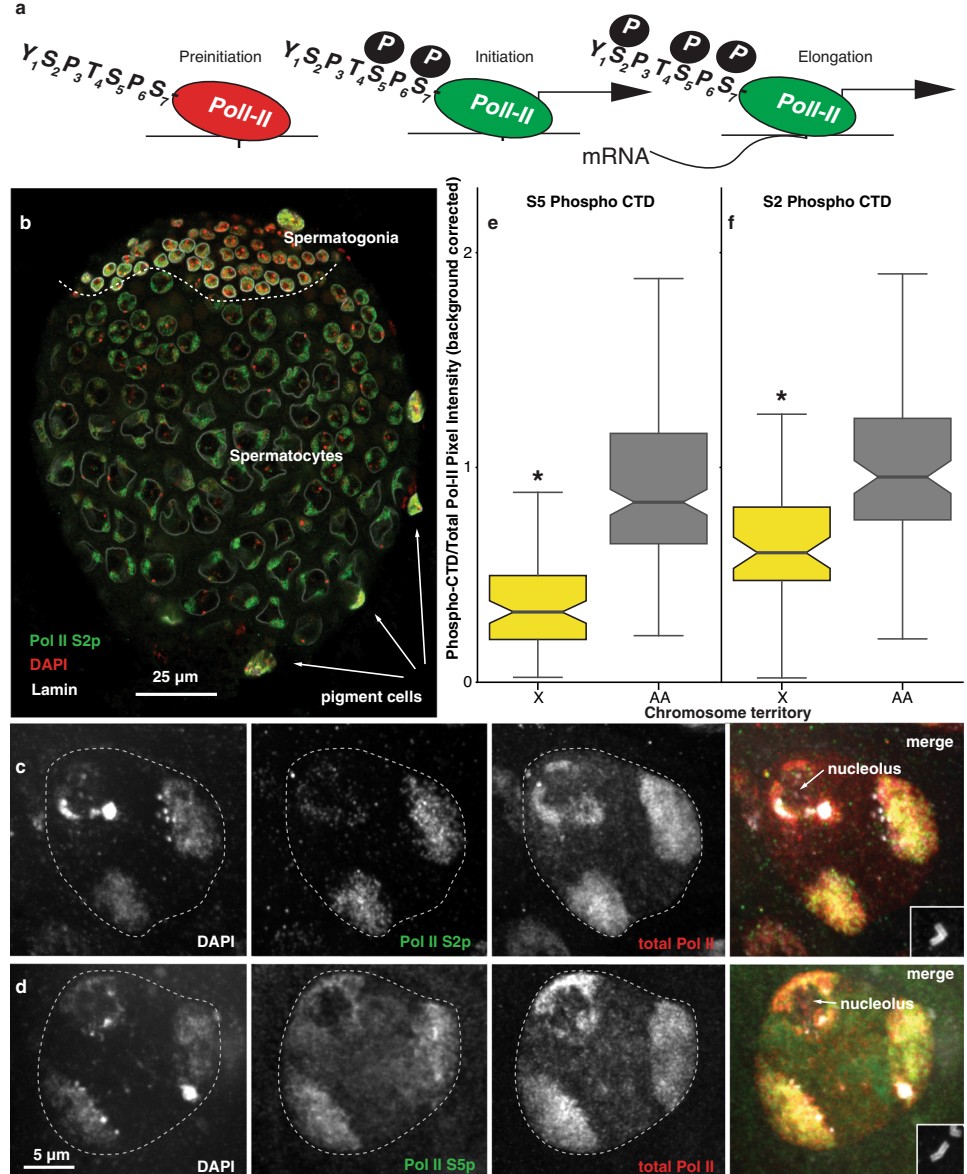

**Fig. 6 Transcriptional events in late primary spermatocyte nuclei. a** Labelled schematic of RNA Pol-II preinitiation, initiation, and elongation at a promoter. Optical sections of an immuno-stained L3 testis (**b**) and spermatocytes (**c, d**). Serine 2 phosphorylated Pol-II localization (green), the nuclear envelope marker Lamin (white), and the DNA stain DAPI (red). We indicate spermatogonia and spermatocytes, separated by a line (dashed, white). Three/five visible pigment cells are indicated. **c** Serine 2 phosphorylated Pol-II localization (green) and total Pol-II (red). **d** Serine 5 phosphorylated Pol-II localization (green) with total Pol-II (red). **c, d** From left to right: DAPI grayscale, Phospho-CTD grayscale, total Pol-II grayscale, color merge. Primary spermatocyte nuclei are outlined (white-dashed line) based on DAPI. Insets in the merged images are anti-Asterless staining (grayscale) used to stage by centriole length, a measure of spermatocyte age. The ratio of **e** Serine 2 phosphorylated Pol-II (261 territories scored) or **f** Serine 5 phosphorylated Pol-II to total Pol-II pixel intensity on the X chromosome and autosome territories ($N = 148$ territories). We corrected for background using non-territory nuclear staining in the same nucleus. Significance ($p \leq 0.01$, two-sided Welch's $t$-test) (asterisk).

(Fig. 6c, d). We also observe localization of total Pol-II, but very little pSer2-CTD on the X territory when we co-stained with X euchromatic oligopaints (Supplementary Fig. 4), consistent with the idea that reduced levels of X chromosome transcripts may be due to decreased active transcription. To obtain more quantitative information on transcriptional status of the X chromosome, we determined the ratio of pSer2-CTD and pSer5-CTD to total Pol-II within a given territory compared to the autosomes in late primary spermatocytes (Fig. 6c, d). We found a > 2-fold decrease in pSer2-CTD ($\log_2 = -1.4$) and a > 2-fold decrease in pSer5-CTD ($\log_2 = -0.6$) on the X relative to the autosomes in late primary spermatocytes (Fig. 6e, f). These data indicate that the decline in X chromosome transcripts seen by scRNA-seq is due to a block in

the transcriptional cycle regulated by CTD tail phosphorylation. Given that the 4th is also in this territory, it may be subject to the same fate.

## Discussion

Our data clearly support a dynamic model, where X chromosomes are expressed at a higher rate in spermatogonia than one would expect based on DNA copy number alone, supporting the idea of X chromosome dosage compensation in the pre-meiotic male germline. This initial upregulation of X chromosome expression is followed by a dramatic decrease. We suggest that lower expression of the X in early meiosis is not due to the

absence of X-chromosome dosage compensation in the germ-line[11], but to an even more extreme reduction in gene expression due to silencing[10,79]. While the canonical dosage compensation pathway, acting to up-regulate X expression, is absent in male germ cells[23,24], there is also evidence for non-canonical dosage compensation in testis[21,27]. The mechanism of germline dosage compensation in *Drosophila* is unknown. Our data provide an important additional argument against failed dosage compensation in the male germline as a cause of reduced X chromosome expression in spermatocytes, based on the fact that the 4th chromosome undergoes a similar dramatic decrease in transcript levels, despite being present in two copies. Silencing of genes in spermatocytes is independent of copy number.

Mechanistically, the reduced expression of the X and 4th chromosomes in spermatocytes correlates with the failure to activate RNA Pol-II. The Y chromosome is concomitantly active. This begs the question, why? This expression pattern could be due to the simple absence of genes expressed in spermatocytes on the X and 4th chromosomes and the presence of genes that must be expressed from the Y chromosome (Fig. 7a). If there are few genes expressed, there will be little active Pol-II ipso facto. However, expression of "housekeeping" genes suggests a chromosome-wide decrease in X and 4th expression. A pure gene content model predicts that genes newly arriving on the X with would be expressed, as evolutionary modification of regulation takes time. In fact, the autosomal *ocnus* gene is precisely expressed in spermatocytes, but shows extremely reduced reporter activity when inserted onto the X[28]. This is consistent with a model where the X is a generally unfavorable environment for spermatocytes gene expression, due to either chromosome- or territory-level repression (Fig. 7b, c). This is reminiscent of meiotic sex chromosome inactivation in mammals, where X chromosome expression is high in spermatogonia, followed by X inactivation associated with a distinct organelle like XY body[15]. The inactivation of both the X and Y chromosomes in mammals may be a special case of a more general inactivation of unpaired chromosome regions in a genomic defense model[16]. Lack of homology could signal intruding transposable elements seeking to hijack the germline for vertical transmission to the next generation. Active recognition and silencing would be useful to the host organism. We observed two violations of the prediction that unpaired chromosomes are silenced in primary spermatocytes. Specifically, the 4th chromosome would be active, and the Y would be inactive in the simplest versions of this model. However, the 4th has retained its X chromosome-like silencing despite having two copies and the Y is maximally expressed in spermatocytes despite having a single copy. One way to achieve this would be the creation of a repressed territory occupied by both the X and 4th chromosomes (Fig. 7c). The evolutionarily retained inactivation of the 4th could be due to this localization, perhaps originally triggered by monosomy in ancestral species. It is also possible that the non-recombining 4th chromosome[4] is not recognized as having a homolog. The single Y is highly diffuse and very little of it is in this repressed territory. However, allele-specific expression of the Y-linked rRNA genes drive the activity of the nucleolus[75], so at least part of the Y is expressed while in a repressed territory. It is possible that Y-linked genes, including the rDNA cluster, required for spermatogenesis escape inactivation as occurs for a subset of X linked genes on inactive X chromosomes in mammals[80]. X to 2nd or 3rd chromosome translocations result in breakpoint-independent dominant male sterility, whereas X to 4th do not[74]. Spreading repression or activation along a chromosome element, or relocation of parts of elements to novel territories might result in such a phenotype. Experiments to test these

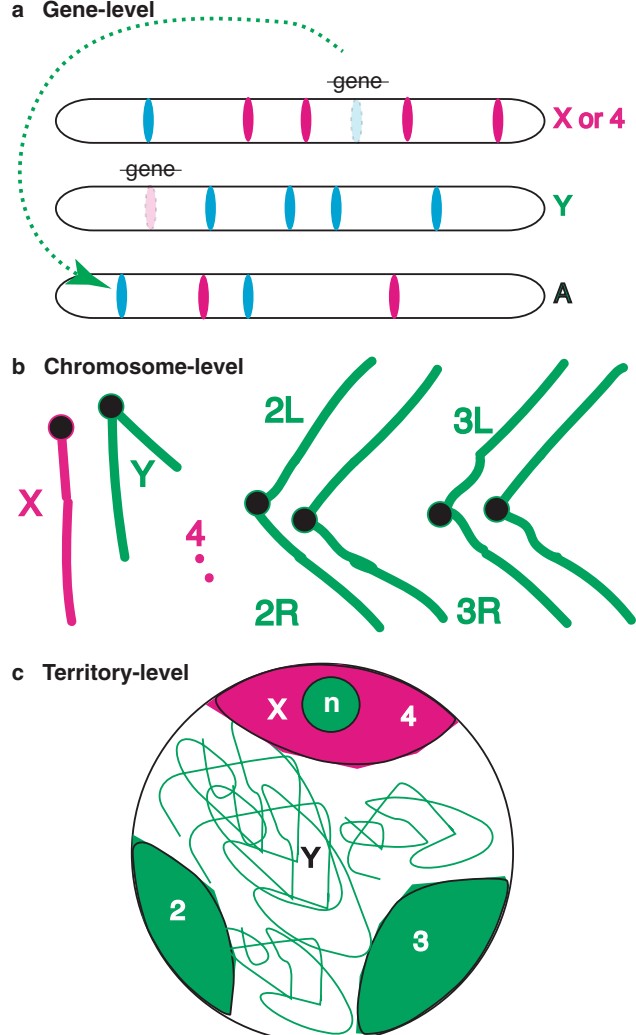

**Fig. 7 Models for chromosome element expression in spermatocytes.** Non-mutually exclusive models. **a** Gene-level regulation. Movement (green dashed arrow) of genes with male-biased expression (blue ovals) off the X due to deleterious changes resulting from antagonistic selection in females or extinction of genes with male-biased expression on the X, or female-biased expression on the Y (strike through) result in a feminized X and a masculinized Y. X and 4th (red), Y (green), and major autosome (black) chromosomes (elongated bars) shown. **b** Chromosome-level regulation. This model relies on chromosome-specific regulation, including X-chromosome inactivation. Gene expression depends on chromosome elements that are active (green chromosomes) or repressed (red). Centromeres (Black circles). **c** Territory-level regulation. In this model the X and 4th chromosomes are restricted to a repressed address (red lozenge) within the nucleus (large circle). This could be the result of a barrier preventing activation of Pol-II or active repression within the territory. The 2nd and 3rd chromosome territories (green lozenges) and the diffuse Y (green lines) are expressed normally. The active nucleolus (n, green circle) is embedded within the repressed domain, but depends on expression of Y rDNA, not X rDNA.

models will help us understand the evolution of sex chromosome expression in flies, and probably many other species.

## Methods

**Sample preparation.** We collected gonads from the immobile third instar (L3) larvae just prior to the final prepupal defecation and prior to eversion of the anterior spiracle (Supplementary Fig. 1a). Staging was aided by staining gut contents using 0.5 mg/ml Sulforhodamine B yeast paste fed to L3 larvae (clear foregut,

full midgut/hindgut) (Supplementary Fig. 1b). We dissected L3 larval and 1–5 days old adult gonads and brains directly into phosphate-buffered saline, pH 7.4 (PBS). We used fixed gonads for immunostaining and gonads and brains for in situ hybridization, and gonads for bulk RNA-Seq and scRNA-seq. For expression profiling whole gonads and single cells, we used enzymatic and mechanical treatments to remove non-gonadal tissue. We prepared frozen single-use aliquots of all enzymes used in preparing gonads/cells and tested to determine optimal digestion times. We removed fat body from gonads (Supplementary Fig. 1c–f) using 0.00075 U/μl Papain and 0.0125 U/μl Collagenase in PBS at 22 °C for 2–3 min in 1.5 ml Eppendorf tubes mixed by gently flicking the tube with a forefinger. We decanted floating fatbody cells and washed 2× with PBS. For whole gonad profiling, we saved the gonads in 350 μl RLT buffer mixed two times for 30 s, froze in a dry ice-ethanol bath and stored at −80 °C. We performed RNA-Seq on eight biological replicates each of cleaned L3 larval testes and ovaries. For single cell preparation, we pre-coated all tubes, pipettes, and filters with 0.5% Bovine Serum Albumin (BSA) to minimize loss of cells. We removed fat body from gonads as above, except we used 0.075% (w/v) porcine powdered pancreas (PPP) rather than papain (Supplementary Fig. 1c, d). We dissociated gonads in 0.45% PPP and 0.23 U/μl Collagenase in PBS at 22 °C for 30 min, teased with tungsten needles and pipetting under the dissecting scope. Digestion was stopped by adding fetal bovine serum (FBS, 1% final v/v) for 2 min. We decanted the cell suspension and a wash of 0.04% BSA onto a 35 μm cell filter. Cells were pelleted at 845 × g for 5 min and resuspended in 25 μl PBS, 0.04% BSA. A 1 μl cell suspension was used to calculate density microscopically. We performed scRNA-Seq on three biological replicates of L3 instar larval testes.

In initial trials, we determined that the somatic support cells that tightly encase the germline were separated and that the multicellular germline cysts were disrupted using gonads from flies expressing reporters (*tj > GFP* or *Vasa-GFP*) (Supplementary Fig. 1g-h). The *tj* gene was highly expressed in a small subset of somatic cyst cells in the testis, these wrap tightly around the germcell clusters, so isolation of single *tj+* cells shows that they were separated from the germ cells (Supplementary Fig. 1g). The *vasa* gene was expressed in germline cells, which dramatically vary in size (Supplementary Fig. 1h). We also stained cells with 1 μg/ml Hoechst to detect nuclei then examined them microscopically to assess cell morphology. We typically found single cells (96.5%), but occasionally detected small aggregates of cells (1%, with 4–8 cells, each 6–8 μm), multinucleated cells (2%, mostly two nuclei spermatocyte cysts) and enucleated cells (0.5%) (*n* = 565 cells). We analyzed the viability of cells in 0.2% Trypan blue in 0.5× PBS, 0.02% BSA. 25–30% of cells took up dye, suggesting 70–75% viable cells prior to microfluidic loading (*n* = 1000 cells) (Supplementary Fig. 1i).

**Immunostaining**. For immunostaining formaldehyde fixed tissues, we used slightly different conditions depending on the antibodies used and location. For protein traps: we fixed in 4% formaldehyde in PBS for 20 min; blocked in 1× PBS plus 0.1% Triton X-100 (PBX), 3%BSA, 0.02% NaN₃ and 2% Normal Goat Serum (NGS) for 30 min; incubated with primary antibodies (1:50 mouse α-Fas3, 1:50 mouse (1B1) hu-li tai shao, and 1:5000 chicken α-GFP in PBX) overnight at 4 °C; washed with 1× PBX; incubated with secondary antibodies (1:200 Alexa Fluor 488 α-chicken, 1:100 Alexa Fluor 568 α-mouse) with 1 μg/μl DAPI at room temperature for 2 h; rinsed twice in 1× PBX, once in 1× PBS; and mounted onto a microscope slide in Vectashield. Images were acquired using Zeiss LSM 800 laser scanning confocal microscopes with 63×/1.4 Oil DIC M27× objective and illumination lasers at 405, 488, and 561 nm. For RNA Pol-II: we fixed in 9% formaldehyde in PBS for 20 mins; washed three times in PBS with 0.3% Triton X-100 (PBST); blocked for >1 h in 5% NGS in PBST at room temperature; incubated in primary antibodies (1:1000 rat α-RNA polymerase II subunit B1, phosphorylated CTD Ser-2, or phosphorylated CTD Ser-5), 1:100 mouse α-lamin, 1:30,000 guinea pig α-Asl, 1:50 mouse α-Fas3, 1:10,000 chicken anti-GFP) in PBX with 5% NGS overnight at 4 °C; washed 3× with PBX for 10 min; incubated in secondary antibodies (1:1000 Alexa Fluor 488 goat α-rat, Alexa Fluor 568 goat α-mouse, and Alexa 647 goat α-guinea pig) at room temperature; counterstained with DAPI at 1:1000 in 5% NGS in PBX for 2–8 h at room temperature; 3× in PBX for 10 min at room temperature and mounted in Aquapolymount under a No. 1.5 coverslip. Images of larval testes were acquired using Nikon Eclipse Ti with a 100×/1.4 NA oil immersion objective, a spinning disc confocal head, a CoolSNAP HQ2 camera and illumination lasers at 405, 491, 561, and 642 nm. The microscope was controlled by and images (Supplementary Fig. 2) were acquired using MetaMorph. All data analysis was performed using ImageJ.

**In situ hybridization**. To detect heterochromatin on X, 4th and Y chromosomes, we synthesized probes (1.688 satellite of X chromosome and AATAT repeats for 4th and AATAC repeats for Y chromosomes) using *D. melanogaster* genomic DNA as a template[72,81]. PCR was performed using 50–100 ng/μl of template DNA with Taq Platinum DNA polymerase following the manufacturer's instructions. Probes were PCR-labeled with digoxigenin-11-dUTP or biotin-14-dUTP during their synthesis at the 5′ end. To detect the euchromatin, we synthesized oligopaint probes using 5' fluorophore labeled and 5' phosphorylated PCR primers, amplified with the following cycles: 95 °C for 5 min; 40 cycles of 95 °C for 30 s, 53 °C for 30 s and 72 °C for 15 s, with a final extension step at 72 °C for 5 min. The PCR product was purified using Zymo spin columns (D4031) and then digested using lambda

exonuclease for 30 min at 37 °C and 10 min at 75 °C. The digested probe products were precipitated using ethanol and quantified using a nanodrop.

We stained mitotic chromosomes to assess probe hybridization (Supplementary Fig. 3). For mitotic metaphase chromosomes, we immersed dissected brains in 0.05% Colchicine in PBS for 20 min, hypotonized in tap water for 15 min, and fixed in 3:1 (v/v) ethanol:acetic acid for 15 min, transferred to 60% acetic acid and squashed on a slide. Fluorescence in situ hybridization for heterochromatic probes was performed according to Pinkel et al.[82] with modifications[83] using mitotic chromosome spreading or spermatocytes from adult testes. Post-hybridization washes were performed as follows: two times in 2× SSC at 42 °C for 5 min, two times in 0.1× SSC at 42 °C for 5 min, one time in 2× SCC at 42 °C for 5 min and finally in 2× SSC at room temperature for 10 min. Probes labeled with biotin-16-dUTP were detected using avidin-FITC conjugate (Fisher Scientific) and probes labeled with digoxigenin-11-dUTP were detected using anti-digoxigenin rhodamine (Roche). The preparations were counterstained using DAPI and mounted in Vectashield. Images from metaphase spreads and adult spermatocytes were obtained using a Zeiss Axiophot 2 microscope equipped with Bright field and epifluorescence optics.

For fluorescence in situ hybridization using oligopaints, we fixed testes in 5% formaldehyde washed in 1X PBX and then 0.3 M NaCl, 30 mM Na Citrate, 1% Tween 20 (2× SSCT), then successively in 20, 40, and 50% formamide in 2X SSCT. Testes were pre-denatured at 37 °C for 4 h, 92 °C for 3 min, and 60 °C for 20 min in 50% formamide in 2× SSCT. We heated 100 pmol of the X-euchromatin primary probe and 200 pmol of the 2L-euchromatin primary probe in probe buffer with RNase A and testes to 91 °C for 3 min and incubated overnight at 37 °C for 18 h on a rocker. We washed testes in 50% formamide + 2× SSCT. Testes were then washed with decreasing concentrations of formamide (50 and 20%) and 2× SSCT. To counterstain with antibodies, we refixed with 16% Formaldehyde, 100% Tergitol, 10× PBS, heptane and MilliQ H20, washed in 1× PBX, and then blocked in a 1× PBX + 1.5% BSA mixture for 1 h. Testes were incubated overnight at 4 °C with: 1:20 mouse α-lamin, 1:20 mouse α-lamin DmO, and 1:10,000 guinea pig α-Asl in PBX. Testes were washed in 1× PBX and then incubated in the secondary antibodies. Testes were mounted onto a microscope slide containing a drop of Vectashield. Images were acquired using Zeiss LSM 800 laser scanning confocal microscope with 63X/1.4 NA oil immersion objective and illumination lasers at 405, 488, 561, and 640 nm.

**Image analysis**. To measure the X and 2 L oligopaint territories, we loaded CZI images into *Imaris*. Centriole elongation is a marker of time during the prolonged G2 phase of primary spermatocyte development, which we scored using the Asterless (Asl) centriole marker. We created X and 2 L oligopaint probe channel masks (*n* = 23 for both X and 2 L) of the pixel intensities in late primary spermatocyte cells. We measured volume and sphericity (shape) of the masks and performed Paired *t*-tests.

For quantitative measurements of Ser-2, Ser-5 Pol-II, and total Pol-II signal, samples were dissected and processed in parallel for each experiment and imaged (at 16 bits) on a single day using identical microscope settings ensuring that all pixel intensities were within the dynamic range of the camera (no more than ¾ of the full dynamic range). We normalized X chromosome territory region of interest to the average total fluorescence intensity of the autosomes for each measurement in late apolar spermatocytes determined from that day's experiment. The data presented are from two independent experiments. To quantify the amount of phosphorylated CTD Ser-2 or Ser-5, total Pol-II and DAPI in the discrete DNA domains within the nuclei of spermatocytes, autosomal or nucleolar associated DNA domains that were spatially separated from other domains (in XY and with no overlap in the Z-slices containing any signal from the cluster) were selected from images of spermatocytes. Z-slices encompassing the domain were then summed and the integrated pixel intensity for this volume was measured. An area of nucleoplasm not containing any other DNA domains was selected and measured for background subtraction. X chromosomal domain was identified by the nucleolus region and the other two were considered as autosomal domains. Ratios between X chromosomal and summed autosomal domains were calculated. We co-stained with antibodies to Pol-II and FISH oligopaints to confirm localization patterns to X and 2 territories (Supplementary Fig. 4).

**RNA-Seq—library preparation**. For bulk sequencing, we added fatbody-free gonads to RNeasy 96 kit 350 μl RLT buffer mixed two times for 30 s, froze in a dry ice-ethanol bath and stored at −80 °C. We extracted total RNA from gonad samples using RNeasy 96 kit following the manufacturer's protocol. We quantified RNA using the Quant-iT RiboGreen quantification kit and used 200 ng of total RNA for library preparation[84]. We spiked in External RNA Control Consortium (ERCC) spike-ins pools 78 A and 78B (transcribed from a certified standard reference) prior to fragmentation. For multiplexing, we used eight different TruSeq v2 kit barcoded adaptors. We sequenced stranded multiplexed libraries using a single-ended 50 bp strategy and generated RNA-Seq profiles of biological quad-ruplicates on a HiSeq Illumina 2500.

For single-cell sequencing, we loaded dissociated cells (6 K for Replicates 1 and 2, 12 K for replicate 3) onto the 10X Chromium system for barcoding and library preparation following the user guide for Single Cell 3′ Reagent Kits v2. We quantified libraries with Quant-iT PicoGreen and confirmed 300–500 bp insert

sizes on a TapeStation 2200. We generated scRNA-Seq profiles in biological triplicate pools on the 10X Chromium System and sequenced (Read1 = 26 bp, Read2 = 98 bp and Readi7 = 8 bp) on a HiSeq Illumina 2500.

**RNA-Seq—data processing**. For the bulk RNA-Seq, we demultiplexed and converted Binary Base Call (BCL) files to FASTQ using bcl2fastq (v2.17.1.14, Illumina, San Diego, CA, USA). We processed FASTQ files through custom RNA-Seq workflow (commit: 0609e5c8752, Supplementary Dataset 6). Briefly, reads are trimmed of Illumina adapters using *cutadapt*[85] with default parameter except for --q 20 and --minimum-length=25. We mapped with *Hisat2*[86] with default parameters except --max-intronlen 300,000 and --known-splicesite-infile using annotated splice sites (FlyBase r6-26). We removed multi-mapping reads and low-quality alignments using *samtools view* -q 20[87]. Gene expression and intergenic expression (FlyBase r6-26) is quantified in a strand-specific manner using *FeatureCount* from the *subread* package[88] with the -s 2 option. The workflow outputs a number of quality control metrics from FastQC, FastQ Screen[89], Picard Mark-Duplicates, and Picard CollectRnaSeqMetrics (Supplementary Dataset 6). In addition, we quantified intergenic expression and ERCC spike-in expression using *FeatureCount*[90]. See Supplemental Dataset 1 for visualization of all analyzed scRNA-Seq data and all code is available at the Zendo repository: https://doi.org/10.5281/zenodo.3973174. Bulk RNA-Seq data is available at the Gene Expression Omnibus (GEO) GSE115478 (larval testes) and GSE115511 (larval ovaries). Bulk adult testes and ovaries are from GSE99574[91], Supplementary Dataset 3.

For scRNA-Seq, we converted BCL files to FASTQ using *cellranger mkfastq* (https://support.10xgenomics.com/single-cell-gene-expression/software/pipelines/latest/what-is-cell-ranger, Supplemental datset 6). We demultiplexed cells, aligned reads, and quantified gene-level expression using *cellranger count* with default parameters. For alignment and quantification, we used the *D. melanogaster* genome assembly (dm6) with gene annotations from FlyBase r6.26. Larval testes scRNA-Seq data is available at the GEO GSE125947.

To quantify how well scRNA-Seq captured the overall expression patterns of the testis, we summed gene-level counts from all cells for each of our three replicates and normalized scRNA-Seq and bulk RNA-Seq libraries using Transcripts Per Million reads +1 (TPM +1). We calculated pairwise Spearman rank correlation coefficients among all samples (Supplementary Dataset 3).

**Sex-biased gene expression**. For sex-biased gene expression, we used publicly available adult RNA-Seq counts from for $w^{1118}$ and *Oregon-R* (GEO: GSE99574). Briefly, counts tables from $w^{1118}$ and *Oregon-R* were downloaded for each tissue. Gene identifiers were mapped to current FlyBase FBgns using the provided annotation files and FlyBase's secondary ID. We estimated differential gene expression by comparing males and females using DESeq2 for each adult tissue and for L3 bulk RNA-Seq generated in this study.

**scRNA-seq cell selection, clustering, and cell type annotation**. Prior to the downstream analysis, we removed cell IDs that were likely to be empty and identified cell IDs that likely contain two or more cells known as multiplets (Supplementary Dataset 2). There are two distinct classes of multiplets: homotypic and heterotypic. Homotypic multiplets contain 2 or more cells from the same cell-type. Homotypic multiplets tend to have high UMI counts and are found at the top of the UMI distribution however, setting an upper UMI threshold may bias against high RNA content cell-types. To identify an optimized upper UMI threshold, we performed a grid search over [4000, 5000, 6000] gene expression thresholds. For each point in the grid search, we clustered cells using *Seurat*[43]. We then compared cluster calls using the adjusted rand index and selected an upper UMI threshold which did not drastically change clustering. We found that cluster calls remained stable using an upper threshold of ≤ 5000 expressed genes, which removed 768 cells that were putative homotypic doublets. Heterotypic multiplets contain 2 or more cells from different cell-types. This leads to a mixture of expression profiles causing cells to look like intermediate cell-types, which are identifiable using in silico mixing of cell-types. We removed 410 cell IDs that behave like the mixture.

We combined all three scRNA-Seq replicates into one dataset using single-cell integration[92] and clustered cells with K-Nearest Neighbors using the 2,000 most variably expressed genes[43]. To determine cluster boundaries, we iteratively tested multiple threshold parameters (range 0.2-1.2), selecting a threshold of 0.3, which gives a set of 10 clusters that clearly separate germline and somatic cyst lineages, and each cluster is represented in all three replicates (Supplementary Dataset 2).

We compared gene expression patterns between scRNA-Seq and manually curated images (43 genes from the literature; 31 genes from this study; Supplementary Dataset 4). We developed an overlap score (0–4) between scRNA-Seq expression and mRNA or protein expression in curated images. A score of 4 indicates a gene showed cell type biased expression (scRNA-Seq) in the exact same cell types as mRNA or protein expression (curated images). Since protein expression may lag transcription, we also gave a 4 when protein expression was later in the specific cell lineage. A score of 3 indicates that the gene is highly expressed, but not cell type biased, in the exact same cell types as mRNA or protein expression. A score of 2 indicates cell type biased gene expression in the same cell lineage as mRNA or protein expression. A score of 1 indicates high gene expression

in the same cell lineage as mRNA or protein expression. Finally, a score of 0 indicates that there is no overlap.

Projecting gene expression onto the UMAP clusters is a visually appealing method to rapidly access where genes are expressed in the testis. We generated these projections of cellular resolution expression for every expressed gene in the *Drosophila* genome (FigShare https://doi.org/10.35092/yhjc.11950746). For each gene we plot a UMAP project colored by Z-score value (−3, 3). The line graph represents normalized gene expression for each cluster.

**scRNA-seq downstream analysis**. We know that the transcriptional profile of the testis is unique and expressed genes are not equally distributed among chromosome elements. Therefore, we wanted to interrogate a multiple gene sets to ensure our observations are unbiased. We created three "housekeeping" genes sets, two based tissue-specific expression in adult tissues and two gene sets based on scRNA-seq gene expression from L3 testis. Using the adult tissue panel described in "Sex-biased gene expression", we compared gene counts across all adult tissues using two tissue-specificity estimates Tau and TSPS. We consider genes with $\tau \le 0.5$ or $TSPS \le 1$ as not tissue specific (i.e., "housekeeping"). We consider genes expressed anywhere in our scRNA-Seq data (expressed) or gene expressed in at least 33% of cells (low cell type specificity, CTSP) which are effectively level genes expressed across multiple cell types.

Expression by chromosome arms is complicated as gene expression of an individual cell is sparse in scRNA-Seq, and simple aggregation of cell counts is not appropriate because missingness is not random[93]. Therefore, we used a cell level permutation test approach for all downstream analysis. Briefly, we calculate each the measure of interest (see below) for each individual cell using all gene sets (see above). For pairwise comparisons, we compare the distributions of cells from two cell types using Mann–Whitney $U$ test. We then permute cell type labels and repeat the Mann–Whitney $U$ test comparisons for 10,000 iterations. We calculate the overall $p$-value based on the number of times the permuted Mann–Whitney $U$ test was more extreme than the original observed test. For mean comparisons, we compare the distributions of cells from a given cell type to a random sample, of the same size, from all cells in the experiment using the Mann–Whitney $U$ test for 10,000 iterations. We calculated the overall p-value as the average number of times the observed distribution was significantly different than the permuted distribution.

We calculate 3 different metrics for each cell used with the described permutation test approach. *Normalized Gene Expression* use the Log10 Transcripts Per Million (TPM) UMI counts. *Male-biased Enrichment Scores* used Gene Set Enrichment Analysis (GSEA) with TPM UMI counts and L3 bulk testis-baised expressed genes to estimate an cell level enrichment score. Finally, we compared X and 4th expression with the major autosomal arms (2nd and 3rd) on an individual cell basis. We calculated the X:A,A, and 4,4:A,A ratios for each cell by taking the total X-linked (or 4th-linked) reads divided by the total autosomal-linked reads (2nd and 3rd) normalized by the number of genes per chromosome using all genes and the three subsets of genes with "housekeeping" characteristics.

We list all genes, strains, genetic reagents, antibodies, sequence based reagents, chemical compounds, commercial assays and kits, large-scale datasets, software and algorithms, and others used in our experiments in the FlyBase provided Author Reagent format (Supplementary Dataset 6).

**Reporting summary**. Further information on research design is available in the Nature Research Reporting Summary linked to this article.

## Data availability

All raw sequencing data has been deposited and is publicly available through the NCBI Gene Expression Omnibus under accession codes "GSE125947", "GSE115511", and "GSE115478"). Projections of the expression of each *Drosophila* gene onto the UMPA can be accessed at the NIH Figshare (https://doi.org/10.35092/yhjc.11950746). All data repository details and links are listed in the FlyBase Author Reagent format (Supplementary Dataset 6). All other relevant data supporting the key findings of this study are available within the article and its Supplementary Information files or from the corresponding author upon reasonable request. Source data are provided with this paper in the Supplementary Data files.

## Code availability

All code is at Zenodo (https://doi.org/10.5281/zenodo.3973174). All software packages and environments used are also listed in the FlyBase Author Reagent format (Supplementary Dataset 6).

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

## Acknowledgements

This work utilized the computational resources of the NIH HPC Biowulf cluster (http://hpc.nih.gov), NCBI and FlyBase databases, and the Bloomington *Drosophila* Stock Center. We thank Bob Johnston for oligopaint probe libraries, Gustavo Kuhn for providing the X chromosome satellite probe, Eduardo Gorab for total RNA Pol-II antibodies, and Veronic Souza for checking the 4th chromosome localization in our experiments. This research was supported in part by the Intramural Research Program of the NIH, The National Institute of Diabetes and Digestive and Kidney Diseases (DK015600) and the National Heart Lung and Blood Institute (HL006126), the National Institute of General Medical Sciences (GM126752 and GM120107), the Eunice Kennedy Shriver National Institute of Child Health and Human Development (HD052937) the São Paulo Research Foundation (JP 2015/20844-4, CEPID 2013/08028-1, PT 2015/16661-1, MS 2017/26609-2, MS 2017/14923-4, DD 2019/15212-0 and DD 2019/14788-5).

## Author contributions

S.M., J.M.F., B.O., M.D.V. and E.M. conceptualized the project; S.M. and J.M.F. developed single cell methodology; J.M.F. developed code; M.A., M.M.L.S.P., C.C.A., D.C.C. M., K.C., Z.D., K.M., C.A.M., O.M.P.G., E.R., M.S. and K.Y. validated results; J.M.F., M.A., B.J.G., C.C.A., and C.A.M. performed formal analysis; S.M., M.A., B.J.G., M.M.L.S. P., C.C.A., D.C.C.M., K.C., S.D., Z.D., K.M., C.A.M., O.M.P.G., E.R., M.S., K.Y., H.Y., H.E.S. and V.S. performed experiments; J.M.F. curated metadata; S.M., J.M.F., B.O., M.D.V. and E.M. wrote the original draft; S.M., J.M.F., M.A., B.O., M.D.V. and E.M. edited; S.M., J.M.F., M.A., C.C.A., B.O. and M.D.V. prepared figures; S.M., H.E.S., V.S., N.R., B.O., M.D.V. and E.M. directed the work; S.M. and B.O. administered the project; N.M.R., B.O., M.D.V., M.A. and E.M. acquired funding.

## Competing interests

The authors declare no competing interests.
