## [Peer Review File · Nature Communications]

REVIEWER COMMENTS

Reviewer #1 (Remarks to the Author):

In this study, the authors use single cell RNA sequencing of the *Drosophila* larval testis to investigate the changes in gene expression on specific chromosomes during spermatogenesis. The scRNA-Seq dataset that they generated will be broadly useful for the community as it provides transcriptional profiles of many distinct cell types and identifies new genes of interest for future study. The methodology and the details of the scRNA-Seq data are clearly described and appropriate. The authors analyze these data to better understand the distribution of gene expression across different chromosomes during spermatogenesis and find that the levels of gene expression on the X chromosome and Chromosome 4 falls and the level of gene expression on the Y increases during maturation of the primary spermatocytes. In addition, they show that these changes in gene expression are correlated with a decrease in RNA polymerase II phosphorylation levels on the X-chromosome. These observations provide new information about the dynamics of gene expression in germ cells, but I think that several points should be addressed before publication.

1. A major potential caveat of this study is that the decreases in average gene expression on the X chromosome and chromosome 4 may be because the genes involved in these stages of spermatocyte differentiation are underrepresented on these chromosomes. Indeed, previous reports from some of the authors show that there is a decreased density of testis-biased genes on the X, as they mention in the text. The authors attempt to correct for this by looking at both all genes expressed by any cell type in their dataset and "widely expressed" genes, defined as genes that are expressed by >33% of cells but the validity of this approach is difficult to assess without additional information. For example, how often are the germ cells of interest among the 33% that express a particular gene in the "widely expressed" gene set? Germ cells enact a very specialized program of differentiation so it is quite possible that they would not express many of the genes in this set. A better approach may be to systematically analyze genes that are expressed in both the germ cells of interest and one or more somatic cell types to avoid this caveat. There may be other ways to do this too, but the current approach does not seem sufficient.
2. The authors should provide more information about how the scRNA-Seq data aligns with the bulk RNAseq data in this study and in previous studies. For example, does the decrease in X-chromosome expression in primary spermatocytes fully account for the dramatic reduction in X-chromosome expression they observe by bulk RNAseq? If not, what other cell types exhibit this phenomenon and how does that fit into their model? Also, the authors mention in the introduction that there is evidence for both partial dosage compensation and partial X-chromosome inactivation in male germ cells (Page 3). Looking specifically at the regions that were found to be upregulated and downregulated in these studies, do the scRNA-Seq data confirm these conclusions (i.e. are the same genes affected)? Since X-chromosome gene expression goes down overall during spermatocyte differentiation, it would seem that the effects of X-chromosome inactivation outweigh the effects of dosage compensation, at least when gene expression across the entire chromosome is averaged together. Is this the case? If so, how would the graphs in Fig. 3 look if one focused solely on the regions of the X-chromosome thought to be subject to dosage compensation or to X-chromosome inactivation?
3. How do the authors reconcile the contrasting evidence that the euchromatic regions of the X have a larger volume, which they state is inconsistent with compaction of the X but are also more spherical than Chr. 2L, which they state is consistent with a role for compaction in the regulation of gene expression on the X?
4. I found the second to last paragraph, where the authors discuss their model and the implications of their data hard to follow. By the phrase "at least partial X chromosome dosage compensation," are the authors referring to the observation that dosage compensation appears to be occurring in spermatogonia but not at later stages, or are they referring to the possibility that some regions of the X chromosome may be upregulated while other regions are not (or both)? Also, the authors seem to be suggesting that Chromosome 4 shows a similar behavior as the X

chromosome because it is derived from an ancient X chromosome, but there are no data in this study to support a causal relationship here. It is an interesting speculation but the authors should discuss more clearly why and how they think that the X and X-derived chromosomes would be subject to this effect specifically in germ cells.

Todd Nystul

Reviewer #2 (Remarks to the Author):

Mahadevaraju et al reported important discoveries regarding the expression patterns of Drosophila genomes, especially in the X-linked and 4th-linked gene expression throughout the spermatogenesis process, using highly resolved single cell RNA-Seq. The finding of decreased X-genes in primary spermatocytes is important, in accordance with the MSCI hypothesis. The 4th-linked gene expression provided further evidence, also suggesting some cis mechanism(s) involved in the reduced expression in the autosome that used to be a sex chromosome. The data also provide clear-cut evidence for the previously arguable dosage compensation hypothesis. Furthermore, their analyses of chromosome structure and transcription activity provided mechanistic evidence for understanding of how the X-inactivation happens, including interesting observations. However, a substantial revision has to be done before acceptance.

I am mainly concerned with their analysis and interpretation of Y-linked genes. The expression of Y chromosome has been taken as a violation to the general prediction of MSCI, which is a major claim as a new discovery. There are two issues with this claim:

1. Firstly, as the Y is a gene-poor chromosome, which is not comparable to the X in gene number. With the data of a small number of Y-linked genes, how can it be called "Y chromosome expression"?

2. Secondly, lines 5-7 in Page 8 stated that "This is likely to occur from expression of a few highly transcriptionally active Y-linked genes". How can "a few" highly expressed genes be able to define as a general feature of the Y chromosome expression? A counter argument to this would be that an inactivated sex chromosome can have a few genes with leaky expression, as an inactivation profile of the human X chromosome showed (Carrel and Willard, 2005. Nature 434: 400-404).

It is clear that the two issues above do not allow a general conclusion of Y chromosomal expression given the reported data. It may help to include noncoding expression in the comparison. The number of Y-linked genes and the names of "a few highly" transcribed genes should be given.

An additional issue:

3. Page 8, paragraph 1 "this decreased expression of 4th chromosome genes cannot be due to loss of dosage compensation. Instead, a gain of inactivation is the simplest explanation. " I agree that the decreased expression cannot be due to loss of dosage compensation. But there is a straightforward reason to doubt the hypothesis of the gain as the simplest explanation because it can be just simply inherited from when the 4th was an ancestral X with an ancestral inactivation.

Minor issues and suggestions:

4. A minor issue that would be easy to fix is that the introduction paragraph in Page 2 stated "Non-mutually exclusive reasons for this reduction include: evolutionary re-localization of genes required in males off the X chromosome (8, 12, 13)". The re-localization of genes is not the reason but a consequence of the X inactivation.

5. Ref 12 is an analysis of the RNA-based duplication, although it was the first paper reported the re-localization pattern. It was generalized to the DNA-based duplication a few years later (Vibrantovski et al, 2007. Genome Research 19, 897-903). This paper should be cited together with 8, 12, 13 to make the point.

6. Page 7, Paragraph 2, "The "Housekeeping genes", a name given to a set of genes based on expression in a wide range of Drosophila tissue (tau and TSPS) (61, 62) is inappropriate for our analysis as it showed poor expression in germ cells." This is a very interesting observation and should be included in Abstract.

7. Page 7, paragraph 3: ". There was a significant and progressive decrease ($P \leq 0.001$) in steady-state...". The legend typed as $P \leq 0.01$. They should be same.

8. Page 19, the last paragraph needs to recompose: the Y chromosome issue above; the last sentence " 'sex chromosome nature' could be a conserved aspect ..." does not mean much.

Reviewer #3 (Remarks to the Author):

In this paper, Mahadevaraju et al performed single-cell RNA-seq of drosophila testes, focusing their analysis on the sex chromosomes X and Y, as well as chromosome 4, an autosome derived from an ancient X chromosome. The study of sex chromosomes is important for understanding evolutionary processes specific to them and how these differ from the autosomes. Sex chromosomes also demonstrate interesting regulatory mechanisms that can serve as a paradigm for understanding gene regulation in general. As well as providing these insights, the field is essential to understanding infertility and sex-biased diseases.

The authors defined testicular cell populations by comparing gene expressions with published spatial expression data. They showed that chromosomes X and 4 have reduced gene expression compared to chromosomes 2 and 3 in primary spermatocytes. Immunofluorescence analysis suggested reduced activation of RNA Polymerase II being the cause of this repression. The paper reports a new resource of single-cell transcriptome in drosophila larval testes. A number of analyses needs to be refined to support authors' conclusions. Comments and suggestions for experiments are listed below.

Major comments:

1) general comment: The novelty and significance of the work need to be clarified more to justify strong impact in the field. I appreciate that this work provides a useful resource for drosophila testis biology, but scRNA-seq of drosophila testes was already done (Witt et al, eLIFE, 2019) and the concept of X chromosome inactivation in male germ cells is not novel (reviewed in Vibrantovski, J Genomics, 2014).

2) page 4, fig 1: What is the significance of comparing gene expressions between testis and ovary? To show the testis-specific inactivation of chromosomes X and 4, other tissues should be included in this analysis.

3) page 7: Discussion of comparison of X chromosome and autosomes:

- It is said that "Expression of the single X chromosome relative to the major autosomes (chromosomes 2 and 3, each present in two copies) is not significantly different in spermatogonia or any of the somatic cell types using either all expressed genes or widely expressed genes (Fig 3A, B)". In this sentence it is not clear what the statistical comparison is relative to. In the figure caption this seems to be described as relative to the average of the somatic cells. Are they saying that gonias are not significantly different to the average of somatic cell types, and that also each

somatic cell type is not significantly different to the average of somatic cell types? This should be explained better. Additionally, I'm not sure that the latter assertion is informative.

- It is said that "There was a significant and progressive decrease ($P \leq 0.001$) in steady-state expression of the X chromosome in early, middle and late primary spermatocytes ($E1^\circ$, $M1^\circ$, and $L1^\circ$)."

Does the statistical test actually show there is a progressive decrease, or is the test just showing that each of the spermatocytes is independently significantly less than the average of the somatic cells? If the former, the testing need to be explained better, if the latter, the assertion needs to be clarified.

- It is said that "Expression of 4th chromosome genes paralleled what was seen for the X. There was a significant and progressive decrease ($P < 0.001$) in steady-state expression levels in $M1^\circ$ and $L1^\circ$ (Fig. 3C, D) compared to expression in spermatogonia." It seems that the tests for the X were performed relative to the somatic cells, not spermatogonia – so if the testing is relative to a different cell type for chr4, there is not a 'parallel' present. However, there is confusion as to what the testing is relative to: the figure caption suggests also that for the 4th chromosome the tests were done relative to the somatic cells – so the mention of significant decrease relative to the spermatogonia here is a confusing one which should be clarified.

4) page 8: It is said that "We observed poor expression of the Y in somatic cells, and increased expression in $E1^\circ$, $M1^\circ$, and $L1^\circ$ primary spermatocytes. This is likely to occur from expression of a few highly transcriptionally active Y-linked genes originally identified by the cytologically visible Y-chromosome loops present at these stages (64, 65)." If the authors have the data available, can they not test this likely scenario and confirm or deny it?

5) page 8: It is said that "The decrease in sex chromosome expression in $M1^\circ$ and $L1^\circ$ did not reflect an overall decrease in total gene expression compared to somatic lineages". However, it has just been shown that Y chromosome expression does not decrease. Previous assertions for chrX and chr4 do not show a decrease in sex chromosome expression, they show a decrease in expression relative to the autosomes, which is a different measure.

6) page 8, fig 4C-F: Overlap with DAPI-dense region doesn't necessarily correlate to inactivation as 2L-euc-positive region is also DAPI-dense (fig 4E). Does any heterochromatin marker protein specifically localise on the chr X/4 territory?

7) page 9, fig 4G-H: Volume should be normalised by chromosome length covered by each probe. Chromosomes 3 and 4 should be included in these data.

Minor comments:

8) general comment: Please insert line numbers to help reviewers to refer points.

9) general comment: Please use colour-blind friendly colouring in figures.

10) general comment: Throughout the paper the 'Seq' in 'scRNA-seq' is capitalised. This is not how it is found in the literature and should be changed to 'seq' to match.

11) general comment: Throughout the paper there is a mixture of 'Fig N' and 'Fig. N'. These should be changed to be consistent throughout.

12) page 3: It is not clear why they are using L3 larvae instead of adults and what benefit/questions this brings. Previous work has done scRNA-seq on adult testis (eg. Witt et al, eLIFE, 2019). Maybe something specific about Drosophila biology makes this important? This should be made clear for readers. Should also perhaps be mentioned in the abstract at least?

13) page 4: "expression of male-specific Y chromosome was highly testis-biased" is an unusual thing to conclude since Y chromosome is only present in testis, there can be no 'bias' relative to

ovaries in the standard sense of the word.

14) page 4: "We identified 18,965 single cells across three biological replicates (Spearman $\rho \geq 0.93$, $P < 0.001$; Table S1)". In this context I do not think it is clear what the Spearman's rank refers to. It sounds like it is somehow related to the number of cells when placed after the current sentence, while examining Table S1 shows it is related to gene expression ranks.

15) page 5: replacing "RNA-seq" with "bulk RNA-seq" would make this explanation clearer, especially when it is mentioned right after scRNA-seq.

16) page 7: "widely expressed genes" are defined as genes expressed in $> 33\%$ of all cells in the single cell data. Is expression of these genes biased to specific cell types?

17) page 7: "Drosophila tissue"  "Drosophila tissues"

18) page 7: "dosage compensation" is used to refer to X upregulation in a number of places throughout the manuscript. The term "dosage compensation" generally is used to refer to both X upregulation and X chromosome inactivation mechanisms, and so care should be taken not to use the generic term in describing just one of the mechanisms it encompasses, particularly when talking about both of them in the same paragraph.

19) page 7: Please define "steady state expression" of chrX. Is "steady state expression" a term used commonly to refer to "expression relative to autosomes"? If so, this is fine. If not, then it could be easily misunderstood that their data is showing the transcription from the X absolutely decreases, when in fact it is the ratio of transcriptional activity between X and autosomes that is shown to be decreasing.

20) page 8, second paragraph: How often does chr 4 localise in the region including chr X? Please add the conclusion of this section.

21) page 8: the sentence "Spermatocyte chromosomes are represented (Fig.4)." does not make sense in isolation, I think this sentence has been accidentally inserted.

22) page 8: I think "X chromatin heterochromatic satellite sequences" should read "X chromosome heterochromatic satellite sequences".

23) page 10: "This suggests that sex chromosome, not copy number, determines activity in primary spermatocytes." This sentence does not make sense in this form. I think the sentiment is "This suggests that some property intrinsic to sex chromosomes modulates their expression in a way independent of copy number"?

24) page 10: the sentence "Where X-like chromosomes are inactivated and the Y-like chromosomes are highly expressed." does not make sense in isolation, I think this sentence has been accidentally inserted.

25) page 10: It is unclear what "sex chromosome nature" means.

26) figure 1:

- X and Y chromosomes should be next to each other on the axis.
- Y axis label should explain better what the measure is (I think 'average gene expression', not just 'expression').
- X axis title of 'chromosome arm' is unsuitable since X and Y are not arms. Change to something like 'scaffold' or 'location'.

27) figure 2:

- D-I: bottom right panel is difficult to read: Having it as a line graph does not make sense as the data is not a series. Bar graphs should be used instead. X axis labelling being only on the last panel makes it hard to read for other panels, if bar chart was used with bars colour-coded to match the cell types as in panel A/B, this may be clearer.

- E: line graph suggests highest expression in Gonial and E1⁰, but IF image seems to show higher expression in M1⁰/L1⁰ (based on the cartoon in panel A)?

28) figure S1: there are no scale bars on panels A and B.

Response to review of:
Dynamic Sex Chromosome Expression in Drosophila Male Germ Cells, Mahadevaraju *et al.*

Nature Communications

We have provided a point-by-point response to the reviewers' comments below. The original
review is in **black** followed by our response in **blue**. We have added line numbers (one of the
suggestions) and all changes in the manuscript text file have been noted by blue text and the
corresponding line numbers are referenced in the response. You will notice the large contribution
of new blue text to the manuscript.

**REVIEWER COMMENTS**

Reviewer #1 (Remarks to the Author):

In this study, the authors use single cell RNA sequencing of the Drosophila larval testis to
investigate the changes in gene expression on specific chromosomes during spermatogenesis.
The scRNA-Seq dataset that they generated will be broadly useful for the community as it
provides transcriptional profiles of many distinct cell types and identifies new genes of interest
for future study. The methodology and the details of the scRNA-Seq data are clearly described
and appropriate. The authors analyze these data to better understand the distribution of gene
expression across different chromosomes during spermatogenesis and find that the levels of gene
expression on the X chromosome and Chromosome 4 falls and the level of gene expression on
the Y increases during maturation of the primary spermatocytes. In addition, they show that these
changes in gene expression are correlated with a decrease in RNA polymerase II phosphorylation
levels on the X-chromosome. These observations provide new information about the dynamics
of gene expression in germ cells, but I think that several points should be addressed before
publication.

We are delighted that the major points we were trying to make seem clear, and endeavored to
make the suggested changes and better explain our choices to *Nature Communications* readers.

Thank you for the questions relating to some important points that we glossed over in the
original submission. You've helped make this a much better paper.

1. A major potential caveat of this study is that the decreases in average gene expression on the
X chromosome and chromosome 4 may be because the genes involved in these stages of
spermatocyte differentiation are underrepresented on these chromosomes. Indeed, previous
reports from some of the authors show that there is a decreased density of testis-biased genes on
the X, as they mention in the text.

This is correct. In fact, it is integral to the nature of our arguments for how gene content arose on
the sex chromosomes in the course of evolution (Rice, Charlesworth, Long, Chung-I Wu, Oliver,

etc labs). The problem with gene content and expression on the sex chromosomes is a classic
causality dilemma. In a nutshell, the model is that antagonistic sexual selection leads to
feminization or demasculinization of X chromosomes by gene extinction or movement, which is
permissive for X inactivation. Chromosome-wide inactivation is without question a strong
selective force that would lead to further movement of genes required in spermatocytes to other
parts of the genome or expressing them precociously prior to inactivation. We did not do a good
enough job setting up this problem in the beginning of the paper. We have added text at the
beginning of the paper that clearly states this problem. We have also added a new model (figure
7) to the end of the paper.

[revised manuscript text omitted]

The authors attempt to correct for this by looking at both all genes expressed by any cell type in
their dataset and “widely expressed” genes, defined as genes that are expressed by >33% of cells
but the validity of this approach is difficult to assess without additional information. For
example, how often are the germ cells of interest among the 33% that express a particular gene in
the “widely expressed” gene set?

Germ cells enact a very specialized program of differentiation so it is quite possible that they
would not express many of the genes in this set. A better approach may be to systematically
analyze genes that are expressed in both the germ cells of interest and one or more somatic cell
types to avoid this caveat. There may be other ways to do this too, but the current approach does
not seem sufficient.

We absolutely wanted to measure gene expression in the germline, so we did check carefully to
ensure that the cell types of special interest expressed the widely expressed genes represented.
Specifically, using widely expressed genes gives particularly good coverage in the germline. For
example, 82% of spermatogonia express a CTSP gene, which is roughly the same as in somatic
cells (52-76% of cells). Briefly, we had done everything suggested above prior to the first
submission. The problem is that we failed to bring the reader along with appropriate text and
figures. It is also important to consider all the genes as well as subsets, which we fear got a bit
lost. We have rearranged this section completely, by first looking at all gene expression in figure
3, which is similar to the previous version, but with the widely expressed genes (now scored as a
Cell Type Specific index CTSP) left out. This allows us to introduce the main result cleanly.
Using all genes is the closest to raw results, showing that no special selection of gene sets is
needed for our conclusions. We then raise the caveat noted by the reviewer, and finally provide
the resolution in a formal analysis. This has resulted in a completely new Figure 4 and
completely new paragraphs in the main text (see below), that highlights the results of using
different metrics for “housekeeping” genes. We go through what functions are encoded by those
genes, where they are expressed in the testis, and how the different gene sets affect the results.
This takes some space, but we think it is well used.

[revised manuscript text omitted]

2. The authors should provide more information about how the scRNA-Seq data aligns with the
bulk RNAseq data in this study and in previous studies. For example, does the decrease in X-
chromosome expression in primary spermatocytes fully account for the dramatic reduction in X-
chromosome expression they observe by bulk RNAseq?

We agree that this should have been clearer. While we did mention the high correlation between
the bulk RNASeq and the sum of the scRNASeq, we did not include any detailed analysis. This
has now been added. In terms of gonads, gene expression in spermatocytes explains the bulk
gonad analysis. We have added two new panels to figure 2 (Fig. 2C,D) and new text.

*Main text line 190: If we captured the majority of the cells and cell types, then we should observe a strong correlation*
*between RNA-seq from the whole organ and the total of all single cells. Indeed, the correlation between replicated*
*bulk RNA-Seq from whole L3 testes and sum of single cells from L3 testes was significant (Fig. 2C; Table S3), indicating*
*that major cell types are well represented in our scRNA-seq dataset. Gene Set Enrichment Analysis GSEA on the ten*
*clusters indicated that genes with male-biased expression in whole gonads are also enriched in germ cells relative to*
*somatic cells in the L3 gonad (compare red and blue plots, Fig. 2D). Thus, the germline is the major contributor to the*
*pattern of sex-biased expression of the X, Y, and 4th chromosomes.*

If not, what other cell types exhibit this phenomenon and how does that fit into their model?

In the past, we and others have observed altered sex chromosome expression in other tissues.
For example, we previously observed modestly reduced expression of the male X in the

remaining carcass of gonadectomized male samples. To address this more directly we have
added an analysis at the tissue-level. This provides the most granular data to date on this problem
(although the Fly Cell Atlas will soon exceed this). These new data (128 RNA-seq samples) on
chromosome element distributions of genes with sex-biased expression are in the completely
new Figure 1. These data nicely show the truly special sex chromosome expression (X, Y and
4th) in the gonads, which we map to the spermatocytes two figures later. There are sex
chromosome effects in other tissues, which is beyond the scope of this manuscript, but the salient
point is that only testis provides us with reduced expression of the X and 4th and increased
expression from the Y.

*Main text line 106: Drosophila have X and Y sex chromosomes, two major autosome pairs and a pair of “dot” 4th*
*chromosomes (Fig. 1A). The Y and 4th chromosomes are gene poor, while the remaining chromosome arms are gene*
*rich (Fig. 1B). To examine sex-biased gene expression patterns, we focused on the distribution of male-biased gene*
*expression across chromosomes or chromosome arms (chromosome elements) for each tissue. We measured adult gene*
*expression (quadruplicates) in the whole body (Fig. 1C) as well as seven tissues (Fig. 1D-J): head, thorax (viscera*
*removed), abdomen (viscera and all reproductive organs removed), viscera (including digestive and excretory organs),*
*reproductive tract (gonads and genitalia removed), terminalia (including genitalia and analia), and gonads in females*
*and males from two strains. We found a significant deviation from random (we use $p \leq 0.01$ throughout this study) in 6*
*sample types, including the whole body, head, thorax, viscera, reproductive tract, and gonad (χ^2 test of independence).*
*To examine which chromosome elements contribute to this non-randomness, we performed a post hoc analysis (χ^2 test*
*for each chromosome element (Table S1). Sex chromosomes and former sex chromosomes are the major contributors*
*to the non-randomness.*

*For X-chromosomes, we observed underrepresentation of male-biased gene expression in the whole body from either of*
*two wildtype strains (Fig. 1C), as previously reported *s*. In heads, we observed a slight enrichment in male-biased gene*
*expression in one strain (Fig. 1D). In contrast, we observed a reduction in male-biased gene expression in the*
*reproductive tract (Fig. 1H). The reproductive tract pattern of X chromosome expression is difficult to explain by*
*absence of germline X chromosome dosage compensation or meiotic sex chromosome inactivation, since there are no*
*germ cells in this tissue. By elimination, this suggests that sexual selection drives gene expression patterns of X-*
*chromosome expression in the reproductive tract. In the gonads, we observed an underrepresentation of male-biased*
*gene expression (Fig. 1J), as previously reported *s*.*

*Males with no Y chromosome are viable, but sterile and the Y chromosome is known to be expressed in spermatocytes*
**s*. However, the tissue-specific Y chromosome gene expression pattern is poorly described. We report that Y-*
*chromosome gene expression was detectable only in whole males and gonads (Fig 1C, J).*

*The 4th chromosome showed a decrease in male-biased gene expression in the whole body in one strain (Fig. 1C), an*
*increase in male-biased gene expression in the thorax in one strain (Fig. 1E), and most strikingly, a decrease in male-*
*biased expression in the gonads of both strains (Fig 1J). As a former X chromosome, 4th chromosome expression in the*
*gonads was especially interesting as it mirrored the X-chromosome underrepresentation of male-biased gene*
*expression. Additionally, and unlike the X chromosome, 4th chromosomes are present in two copies in males. Because*
*there are two copies of the 4th chromosome genes, under-representation of male-biased expression cannot be explained*
*by the absence dosage compensation. In summary, only gonads show sex-biased expression of the Drosophila X, Y, and*
*4th chromosomes.*

Also, the authors mention in the introduction that there is evidence for both partial dosage
compensation and partial X-chromosome inactivation in male germ cells (Page 3). Looking
specifically at the regions that were found to be upregulated and downregulated in these studies,
do the scRNA-Seq data confirm these conclusions (i.e. are the same genes affected)? Since X-
chromosome gene expression goes down overall during spermatocyte differentiation, it would
seem that the effects of X-chromosome inactivation outweigh the effects of dosage
compensation, at least when gene expression across the entire chromosome is averaged together.
Is this the case? If so, how would the graphs in Fig. 3 look if one focused solely on the regions of

the X-chromosome thought to be subject to dosage compensation or to X-chromosome
inactivation?

We did not explain this well. Dosage compensation and inactivation are separated by time, not
location on the X chromosome. We have purged the word “partial”, and state more definitively
and plainly, that spermatogonia show dosage compensation and spermatocytes show
inactivation. We previously looked very carefully at whether there were X chromosome regions
with dosage compensation (Gupta et al 2003), and assure the reviewers and editor that we did
carefully look for regional differences along the X, both in terms of upregulation in
spermatogonia and inactivation in spermatocytes in this study, but failed to find any overt
patterns for immediate follow-up. Below is an example of a read pile-up showing reads from the
scRNA-seq distributed along the entire genome. There are no obvious patterns of regional
dosage compensation.

3. How do the authors reconcile the contrasting evidence that the euchromatic regions of the X
have a larger volume, which they state is inconsistent with compaction of the X but are also more
spherical than Chr. 2L, which they state is consistent with a role for compaction in the regulation
of gene expression on the X?

There are only a few papers on Drosophila X chromosome compaction in spermatocytes, and
these were often described rather than shown, or illustrated using camera lucida methods 70+
272 years ago. The idea that there is precocious condensation of the X exists in literature, but has not
been examined carefully using modern methods. We felt like this should be in the paper, even
though the conclusions are ambiguous. We are prepared to remove these data if the editor and
reviewers disagree.

The normalization statement was garbled, which may contribute to misunderstanding. There is
some compaction of the X, but this disappears after correcting for the copy number of 2L (divide
by 2). We now show pre- and post-copy number correction in a new panel in the new Figure 5.
We have rewritten the corresponding text and added blunt statements about failed reconciliation,
as well as our uncertainty about the validity of the copy number correction. We have also
expanded the sphericity text. This measurement is volume corrected by nature: $\phi =$
$((\pi^{1/3})(6\text{Volume})^{2/3})/(\text{Area})$

There is clearly a lot of future work needed, including by genome-wide accessibility and ChiP type experiments. Unfortunately, our access to the imaging core has been limited by the covid pandemic. The additional sequencing is beyond the scope of this manuscript. We are confident in the data we present and believe that these are useful for the community.

Main text line 375: The locations of the satellite sequences identify the territories, but we were most interested in chromosome structure in gene rich euchromatic regions. Hence, we used oligopainting to examine the euchromatic portions of the X chromosome for evidence of compaction that might accompany inactivation. Oligopaint probes show that inactive X chromosomes have greater compaction (decreased volume) and increased sphericity compared to active X chromosomes in mammalian cells 22,26, so we measured both these parameters. We probed similar sized euchromatic regions of the X chromosome (22.3 Mb) and the left arm of the 2nd chromosome (2L, 22.7 Mb) with oligopaints (Fig. 5E). We converted raw in situ data in Imaris to create masks (n=23) of pixel intensities (Movie1) and obtained volumetric measurements of the X and 2 territories (Fig. 5G). We found a that probe length corrected X chromosome volume was reduced relative to chromosome 2L suggesting (Fig. 5H). This was not significant at the $p \leq 0.01$ level we have used in this work ($p = 0.03$). However, when we corrected the data to account for 2L copy number (divided by two), the X had significantly greater volume than 2L (Fig. 5I) which was inconsistent with inactivity resulting from compaction. The assumption that volume scales with copy number is of dubious validity. Interestingly, the X chromosome was significantly more spherical than 2L (Fig. 5J), which was consistent with the hypothesis that compaction accompanies inactivation for regulation of X expression in Drosophila primary spermatocytes. In mammals, the inactive X has a sphericity (ψ) of 0.67, while the active X has $\psi = 0.57$ 22,26. We found that the Drosophila spermatocyte X had $\psi = 0.58$, while 2L had $\psi = 0.53$. Collectively, these results do not provide strong evidence that Drosophila spermatocyte X chromosome activity is regulated by overall condensation levels.

4. I found the second to last paragraph, where the authors discuss their model and the implications of their data hard to follow. By the phrase “at least partial X chromosome dosage compensation,” are the authors referring to the observation that dosage compensation appears to be occurring in spermatogonia but not at later stages, or are they referring to the possibility that some regions of the X chromosome may be upregulated while other regions are not (or both)?

Yes, we were referring to the temporal switch in the lineage between dosage compensation in spermatogonia, by upregulation of the X followed, by X inactivation in spermatocytes. As outlined above, we have dropped the misleading term “partial”, which already helps. We have rewritten this paragraph to emphasize the dynamic change over time/stage.

Main text line 416: Our data clearly support a dynamic model, where X chromosomes are expressed at a higher rate in spermatogonia than one would expect based on DNA copy number alone, supporting the idea of X chromosome dosage compensation in the pre-meiotic male germline. This initial up-regulation of X chromosome expression is followed by a dramatic decrease. We suggest that lower expression of the X in early meiosis is not due to the absence of X-chromosome dosage compensation in the germline 11, but to an even more extreme reduction in gene expression due to silencing 10,29. While the canonical dosage compensation pathway, acting to up-regulate X expression, is absent in male germ cells 23,24, there is also evidence for non-canonical dosage compensation in testis 21,27. The mechanism of germline dosage compensation in Drosophila is unknown. Our data provides an important new argument against failed dosage compensation in the male germline as a cause of reduced X chromosome expression in spermatocytes, based on the fact that the 4th chromosome undergoes a similar dramatic decrease in transcript levels, despite being present in two copies. Silencing of genes in spermatocytes is independent of copy number.

Also, the authors seem to be suggesting that Chromosome 4 shows a similar behavior as the X chromosome because it is derived from an ancient X chromosome, but there are no data in this study to support a causal relationship here. It is an interesting speculation but the authors should discuss more clearly why and how they think that the X and X-derived chromosomes would be

subject to this effect specifically in germ cells.

That the 4th was once an X chromosome is based on the literature (especially from the Bachtrog
lab), and is reasonably established. That the 4th chromosome shows X-like expression is new and
was poorly explained. Causality is often speculative and our conclusions can be labelled as such,
but the logic is reasoned. We have explained that logic more extensively. This has resulted in
new text in several places. There is new introductory text, and the completely new Figure 1A,B.
There is also new text on the 4th related to the new bulk RNA-seq.

*Main text line 106: Drosophila have X and Y sex chromosomes, two major autosome pairs and a pair of “dot” 4th*
*chromosomes (Fig. 1A). The Y and 4th chromosomes are gene poor, while the remaining chromosome arms are gene*
*rich (Fig. 1B).*

*Main text line 134: The 4th chromosome showed a decrease in male-biased gene expression in the whole body in one*
*strain (Fig. 1C), an increase in male-biased gene expression in the thorax in one strain (Fig. 1E), and most strikingly,*
*a decrease in male-biased expression in the gonads of both strains (Fig 1J). As a former X chromosome, 4th*
*chromosome expression in the gonads was especially interesting as it mirrored the X-chromosome underrepresentation*
of male-biased gene expression.

We have data in Figures 3 and 4 showing that the X and 4th have a similar pattern of expression
during male germline development. We have tried to be clearer in discussing these patterns.

354

[revised manuscript text omitted]

Reviewer #2 (Remarks to the Author):

Mahadevaraju et al reported important discoveries regarding the expression patterns of
Drosophila genomes, especially in the X-linked and 4th-linked gene expression throughout the
spermatogenesis process, using highly resolved single cell RNA-Seq. The finding of decreased
X-genes in primary spermatocytes is important, in accordance with the MSC1 hypothesis. The
4th-linked gene expression provided further evidence, also suggesting some cis mechanism(s)
involved in the reduced expression in the autosome that used to be a sex chromosome. The data
also provide clear-cut evidence for the previously arguable dosage compensation hypothesis.
Furthermore, their analyses of chromosome structure and transcription activity provided
mechanistic evidence for understanding of how the X-inactivation happens, including interesting
observations. However, a substantial revision has to be done before acceptance.

Thank you. This is an outstanding summary of what we hoped to convey in this manuscript. We
appreciate critical feedback and are happy to have the opportunity to make improvements.

I am mainly concerned with their analysis and interpretation of Y-linked genes. The expression
of Y chromosome has been taken as a violation to the general prediction of MSC1, which is a
major claim as a new discovery. There are two issues with this claim:

1. Firstly, as the Y is a gene-poor chromosome, which is not comparable to the X in gene
number. With the data of a small number of Y-linked genes, how can it be called “Y
chromosome expression”?

The Y is gene poor relative to the X. We found expression of 2,207 genes from the X and 42
 from the Y. It was a mistake to not specify how many genes are on these chromosomes and how
 many were expressed in the main text – the reader should not have to dig through the supplement
 for this. We also failed to emphasize that the Y is not small. In mitotic chromosomes, the Y is
 actually larger than the X. We take care of this introduction in the first panels of the new Figure
 1 (Fig. 1A,B). We provide numbers again with a new table (Table 1) to support the New Figures
 3 and 4.

 *Main text line 106: Drosophila have X and Y sex chromosomes, two major autosome pairs and a pair of “dot” 4th*
 *chromosomes (Fig. 1A). The Y and 4th chromosomes are gene poor, while the remaining chromosome arms are gene*
 *rich (Fig. 1B).*

 *Main text line 676: Table 1*

Table 1. Chromosome element genes and expression.

Chromosome Element	Annotated ^a	Expressed ^b	CTSP ^c	τ _α ^d	TSPS ^e
X	2,675	2,207	65	336	1,130
Y	113	42	0	0	0
2L	3,501	2,767	126	367	1,252
2R	3,628	2,861	138	406	1,413
3L	3,466	2,867	101	371	1,347
3R	4,201	3,467	145	498	1,673
4	111	102	3	7	54

 We also have new text on Y expression in the new figure 1.

 *Main text line 130: Males with no Y chromosome are viable, but sterile and the Y chromosome is known to be*
 *expressed in spermatocytes ³⁰. However, the tissue-specific Y chromosome gene expression pattern is poorly described.*
 *We report that Y-chromosome gene expression was detectable only in whole males and gonads (Fig 1C, J).*

 We also failed to emphasize that the diffuse nature of the Y in the region between the major
 DNA dense territories in spermatocytes. The reason for diffuse Y chromatin (much more diffuse
 than any other chromosome) and the high overall expression is the shockingly large genes with
 megabase sized introns. This is now added to the text.

 *Main text line 359: The nuclear interior, more diffusely stained by DAPI, was occupied by the large transcriptionally*
 *active Y chromosome ^{66,67,22} (Fig. 5A). The Y chromosome expressed only 42 genes in our experiments, but because of*
 *their megabase introns ²³, this represents extensive transcription along the length of the chromosome.*

 More to the point, we now fully discuss gene-centric, chromosome-centric, and territory-centric
 models for inactivation at the end of the paper.

 *Main text line 430: Mechanistically, the reduced expression of the X and 4th chromosomes in spermatocytes correlates*
 *with the failure to activate RNA Pol-II. The Y chromosome is concomitantly active. This begs the question, why? This*
 *expression pattern could be due to the simple absence of genes expressed in spermatocytes on the X and 4th*
 *chromosomes and the presence of genes that must be expressed from the Y chromosome (Fig 7A). If there are few*
 *genes expressed, there will be little active Pol-II ipso facto. However, expression of “housekeeping” genes suggests a*
 *chromosome-wide decrease in X and 4th expression. A prediction of a pure gene content model is that genes newly*
 *arriving on the X with would be expressed, as evolutionary modification of regulation takes time. In fact, the autosomal*
 *ocnus gene is precisely expressed in spermatocytes, but shows extremely reduced reporter activity when inserted onto*
 *the X ²⁸. This is consistent with a model where the X is a generally unfavorable environment for spermatocytes gene*
 *expression, due to either chromosome- or territory-level repression (Fig. 7B,C). This is reminiscent of meiotic sex*

*chromosome in mammals, where X chromosome expression is high in spermatogonia, followed by X inactivation*
*associated with a distinct organelle like XY body¹⁵. The inactivation of both the X and Y chromosomes in mammals may*
*be a special case of a more general inactivation of unpaired chromosome regions in a genomic defense model¹⁶. Lack*
*of homology could signal intruding transposable elements seeking to hijack the germline for vertical transmission to*
*the next generation. Active recognition and silencing would be useful to the host organism. We observed two violations*
*of the prediction that unpaired chromosomes are silenced in primary spermatocytes. Specifically, the 4th chromosome*
*would be active, and the Y would be inactive in the simplest versions of this model. However, the 4th has retained its X*
*chromosome-like silencing despite having two copies and the Y is maximally expressed in spermatocytes despite having*
*a single copy. One way to achieve this would be the creation of a repressed territory occupied by both the X and 4th*
*chromosomes (Fig. 7C). The evolutionarily retained inactivation of the 4th could be due to this localization, perhaps*
*originally triggered by monosomy in ancestral species. It is also possible that the non-recombining 4th chromosome is*
*not recognized as having a homolog. The single Y is highly diffuse and very little of it is in this repressed territory.*
*However, allele-specific expression of the Y-linked rRNA genes drive the activity of the nucleolus²⁵, so at least part of*
*the Y is expressed while in a repressed territory. It is possible that Y-linked genes, including the rDNA cluster, required*
*for spermatogenesis escape inactivation as occurs for a subset of X linked genes on inactive X chromosomes in*
*mammals⁸². Interestingly, X to 2nd or 3rd chromosome translocations result in breakpoint-independent dominant male*
*sterility, whereas X to 4th do not²⁴. Spreading repression or activation along a chromosome element, or relocation of*
*parts of elements to novel territories might result in such a phenotype. Experiments to test these models will help us*
*understand the evolution of sex chromosome expression in flies, and probably many other species.*

2. Secondly, lines 5-7 in Page 8 stated that “This is likely to occur from expression of a few
highly transcriptionally active Y-linked genes”. How can “a few” highly expressed genes be able
to define as a general feature of the Y chromosome expression? A counter argument to this
would be that an inactivated sex chromosome can have a few genes with leaky expression, as an
inactivation profile of the human X chromosome showed (Carrel and Willard, 2005. Nature 434:
400-404).

We agree with the reviewer, who raises an excellent point. We have changed the text
accordingly. The combination of Muller’s ratchet and selection for a few critical genes that
escaped inactivation could easily result in exactly what we observe. This is an attractive
possibility that can help explain results from multiple species with a single model. We have
added this and the suggested reference to the manuscript in text discussing the possible
contribution of pairing to the Y pattern. This is near the end of the paper, where we discuss
models.

*Main text line 458: It is possible that Y-linked genes, including the rDNA cluster, required for spermatogenesis escape*
*inactivation as occurs for a subset of X linked genes on inactive X chromosomes in mammals⁸².*

It is clear that the two issues above do not allow a general conclusion of Y chromosomal
expression given the reported data.

As outlined above, we have now addressed the facts of Y chromosome expression and included
possible models for Y chromosome expression that include gene content, lack or silencing, and
escape from inactivation models.

It may help to include noncoding expression in the comparison.

We agree and we did not exclude non-coding genes in the original submission. All genes
included non-coding ones throughout the manuscript. We now mention this explicitly at the
beginning in legend of Figure 1.

Table 1 footnote, line 678: a Genes annotated in FlyBase r6.26, including non-coding genes.

Legends text line 687: (B) Haploid annotated gene content of chromosome elements (including non-coding genes).

The number of Y-linked genes and the names of “a few highly” transcribed genes should be given.

We agree. Again, the vague mention of “a few highly” expressed genes was an unfortunate word choice that belies the fact that the Y and 4th chromosomes have similar gene content (but vast physical size differences). We have mentioned the inclusion of Fig 1A,B and Table 1, which provides the reader with important background and new data on Y chromosome gene content and expression. We now include the number of expressed Y chromosome genes in the main text as well.

Table 1. Chromosome element genes and expression.

Chromosome Element	Annotated ^a	Expressed ^b	CTSP ^c	Tau ^d	TSPS ^e
X	2,675	2,207	65	336	1,130
Y	113	42	0	0	0
2L	3,501	2,767	126	367	1,252
2R	3,628	2,861	138	406	1,413
3L	3,466	2,867	101	371	1,347
3R	4,201	3,467	145	498	1,673
4	111	102	3	7	54

Main text line 287: We observed poor expression of the Y in somatic cells and spermatogonia, and increased expression in primary spermatocytes (E1°, M1°, and L1°), concomitant with decreased expression of the X and 4th chromosomes. This occurs from expression of a 42 transcriptionally active Y-linked genes, consistent with the diffuse chromatin and Y-loops originally identified by cytology of primary spermatocytes 66.67.

Main text line 360: The Y chromosome expressed only 42 genes in our experiments, but because of their megabase introns 23, this represents extensive transcription along the length of the chromosome.

The expressed Y genes were: *kl-2*, *ORY*, *Ppr-Y*, *Pp1-Y2*, *ARY*, *CR40441*, *CR41423*, *Su(Ste):CR41533*, *CR41506*, *CR41507*, *CR41509*, *CR42201*, *Su(Ste):CR42407*, *Su(Ste):CR42410*, *Su(Ste):CR42412*, *Su(Ste):CR42414*, *Su(Ste):CR42416*, *Su(Ste):CR42420*, *Su(Ste):CR42422*, *Su(Ste):CR42425*, *Su(Ste):CR42426*, *Su(Ste):CR42427*, *Su(Ste):CR42429*, *Su(Ste):CR42430*, *Su(Ste):CR42432*, *Pp1-Y1*, *CR43176*, *FDY*, *CG45765*, *CR45771*, *CR45775*, *CR45780*, *kl-3*, *kl-5*, *Su(Ste):CR45796*, *WDY*, *PRY*, *Mst77Y-7*, *CCY*, *CR46150*, *CR46158*, *CR46160*, *CR46161*, *CR46165*, *CR46167*, *CR46170*, *CR46178*, *CR46182*, *CR46185*, *CR46187*, *CR46188*, *CR46190*, *CG46191*, *CG46192*, *CR46279*.

This is a long list of coding and non-coding genes (the “CR” prefix denotes non- or minimally encoding genes with short ORFs of unknown function). We are reluctant to include just the names of the Y genes in the main text, as this would create an asymmetry, by leaving out the X and 4th. Including the X and 4th gene names in the main text would take a lot of space (>2 K names) and not contribute much to the argument. Information on expressed genes is found in the supplement, including a specific sheet for the Y chromosome (component of **Table S2**) and aficionados have access to all the data, both in the supplement and in the GEO entries. The reader can also download **File S1** from the NIH Figshare

(<https://doi.org/10.35092/yhjc.11950746>), and see the projection of each Y chromosome gene on
the Figure 2 UMAPs (all genes in the genome are included).

An additional issue:

3. Page 8, paragraph 1 “this decreased expression of 4th chromosome genes cannot be due to loss
of dosage compensation. Instead, a gain of inactivation is the simplest explanation.” I agree that
the decreased expression cannot be due to loss of dosage compensation. But there is a
straightforward reason to doubt the hypothesis of the gain as the simplest explanation because it
can be just simply inherited from when the 4th was an ancestral X with an ancestral inactivation.

We agree completely. We were a little too focused on emphasizing that the X and 4th were
inactivated, not simply de-compensated. A result was poor phrasing. There are two timeframes
to consider. One is developmental time, where there is a clear gain of inactivation early in the 3
587 day long primary spermatocyte stage. The other is evolutionary time, where the reason for 4th
inactivation in development is likely to inherited inactivation from when the 4th was an X. So,
both “gain-of-activation” and “retained X-inactivation” are valid, depending on the timeframe.

*Main text line 277: Expression of 4th chromosome genes parallels what was seen for the X (Fig. 3B). The expression*
*ratio of the two 4th chromosomes relative to the two sets of major autosomes hovered near 1. There was a significant*
*decrease in relative 4th chromosome expression in middle and late primary spermatocytes (M1°, and L1°) compared to*
*expression in either spermatogonia or somatic cells. Since we can rule out failed dosage compensation as a cause of 4th*
*chromosome decreased expression, there must be a gain of inactivation during the developmental transition from*
*mitotic spermatogonia to meiotic spermatocytes. This X chromosome like behavior may reflect the evolutionary history*
*of the 4th chromosome; specifically, that the 4th retained X-inactivation after reacquiring autosomal status.*

Minor issues and suggestions:

4. A minor issue that would be easy to fix is that the introduction paragraph in Page 2 stated
“Non-mutually exclusive reasons for this reduction include: evolutionary re-localization of
genes required in males off the X chromosome (8, 12, 13)”. The re-localization of genes is not
the reason but a consequence of the X inactivation.

We agree with the reviewer, but actually think this is a major rather than a minor point. It is a
classic causality dilemma, and another case of over-simplification in our introduction.
Detrimental male expression of X linked genes with female fitness advantages, would lead to
repression or inactivation. If enough of the genes expressed in spermatocytes have decamped the
X, then we could confuse lack of transcription (a passive state) with inactivation (active
repression). We have written a new paragraph in the introduction that outlines the causality
dilemma.

*Main Text Line 86: Given the dramatic differences in the gonads and gametes between the sexes, the optimal male and*
*the optimal female genome will differ. For autosomes, which reside in each sex in equal dose, selection is balanced. In*
*stark contrast, sex chromosome residency is not balanced. In a population with equal numbers of males and females,*
*2/3rds of X chromosomes reside in females. The X chromosome residency profile is expected to result in more*
*opportunities for selection of alleles favoring females. The Y, of course, resides only in males and is under selection*
*only in males. The presence of a homolog is also important. The single X and single Y chromosomes in males are*
*under immediate selection, while only alleles with some degree of dominance are immediately selected in females.*
*Assuming that there is at least subtle dominance 29, then the X chromosome should be both feminized and*
*demasculinized (alleles with female advantage selected for, and alleles with male advantage selected against), and the*

*Y should be both masculinized and defeminized* 8,13. These patterns have been observed in *Drosophila* species, where
expression from the X chromosome is reduced, and where genes required in males have evolutionarily relocated to
other chromosomes 12. Evolutionary arguments for sex chromosome gene content and expression present an
interesting causality dilemma. In the model, antagonistic selection for female functions on the X drives removal of
genes that males need for development from the X. Removal of those genes is permissive for events such as X
inactivation in the male germline18. It then follows that X inactivation in the male germline would provide even more
selective pressure against X genes with male-biased functions. In this work, we ask if sex chromosome expression is
dynamic at tissue and single cell resolutions.

Interestingly, the Parsch lab’s transgenic experiments showing that a gene with spermatocyte-
specific expression is inactive when inserted anywhere on the X, indicates that there is a
chromosome-wide repression. This puts genes that are required in males under ongoing pressure
to move as a consequence. Cause and consequence are in a loop. and the discussion of passive
inactivity at the gene-level versus active repression at the chromosome-level is included in the
text accompanying the new model figure 7.

*Main text line 430: Mechanistically, the reduced expression of the X and 4th chromosomes in spermatocytes correlates*
*with the failure to activate RNA Pol-II. The Y chromosome is concomitantly active. This beg the question, why? This*
*expression pattern could be due to the simple absence of genes expressed in spermatocytes on the X and 4th*
*chromosomes and the presence of genes that must be expressed from the Y chromosome (Fig 7A). If there are few*
*genes expressed, there will be little active Pol-II ipso facto. However, expression of “housekeeping” genes suggests a*
*chromosome-wide decrease in X and 4th expression. A prediction of a pure gene content model is that genes newly*
*arriving on the X with would be expressed, as evolutionary modification of regulation takes time. In fact, the autosomal*
*ocnus gene is precisely expressed in spermatocytes, but shows extremely reduced reporter activity when inserted onto*
*the X 28. This is consistent with a model where the X is a generally unfavorable environment for spermatocytes gene*
*expression, due to either chromosome- or territory-level repression (Fig. 7B,C). This is reminiscent of meiotic sex*
*chromosome in mammals, where X chromosome expression is high in spermatogonia, followed by X inactivation*
*associated with a distinct organelle like XY body15. The inactivation of both the X and Y chromosomes in mammals may*
*be a special case of a more general inactivation of unpaired chromosome regions in a genomic defense model 16. Lack*
*of homology could signal intruding transposable elements seeking to hijack the germline for vertical transmission to*
*the next generation. Active recognition and silencing would be useful to the host organism. We observed two violations*
*of the prediction that unpaired chromosomes are silenced in primary spermatocytes. Specifically, the 4th chromosome*
*would be active, and the Y would be inactive in the simplest versions of this model. However, the 4th has retained its X*
*chromosome-like silencing despite having two copies and the Y is maximally expressed in spermatocytes despite having*
*a single copy. One way to achieve this would be the creation of a repressed territory occupied by both the X and 4th*
*chromosomes (Fig. 7C). The evolutionarily retained inactivation of the 4th could be due to this localization, perhaps*
*originally triggered by monosomy in ancestral species. It is also possible that the non-recombining 4th chromosome 4 is*
*not recognized as having a homolog. The single Y is highly diffuse and very little of it is in this repressed territory.*
*However, allele-specific expression of the Y-linked rRNA genes drive the activity of the nucleolus 25, so at least part of*
*the Y is expressed while in a repressed territory. It is possible that Y-linked genes, including the rDNA cluster, required*
*for spermatogenesis escape inactivation as occurs for a subset of X linked genes on inactive X chromosomes in*
*mammals 82. Interestingly, X to 2nd or 3rd chromosome translocations result in breakpoint-independent dominant male*
*sterility, whereas X to 4th do not 74. Spreading repression or activation along a chromosome element, or relocation of*
*parts of elements to novel territories might result in such a phenotype. Experiments to test these models will help us*
*understand the evolution of sex chromosome expression in flies, and probably many other species.*

5. Ref 12 is an analysis of the RNA-based duplication, although it was the first paper reported
the re-localization pattern. It was generalized to the DNA-based duplication a few years later
(Vibrantovski et al, 2007. Genome Research 19, 897-903). This paper should be cited together
with 8, 12, 13 to make the point.

Agreed and done. Thank you for suggesting a self-citation!

6. Page 7, Paragraph 2, “The “Housekeeping genes”, a name given to a set of genes based on

expression in a wide range of Drosophila tissue (tau and TSPS) (61, 62) is inappropriate for our
analysis as it showed poor expression in germ cells.” This is a very interesting observation and
should be included in Abstract.

This is another minor point that we feel is important. Reveiwer #1 also brought this up. There is
a new figure 4 and substantial new text (see response to reviewer #1, major point #1). We chose
to explain why we did what we did, rather than explaining why we had to devise a new list of
commonly expressed genes. Tau and TSPS are now in the new figure so that we can discuss
why we felt that we needed another metric. This might ultimately help resolve the issue of the
absence of dosage compensation in the testis reported by the Presgrave lab which is a well-
known and controversial paper in the field. We have also appended the positive data on widely
expressed genes in the abstract.

*Abstract text Line 39: Using single cell RNA-Seq on larvae, we demonstrate that the single X and pair of 4th*
*chromosomes are specifically inactivated in primary spermatocytes, based on measuring all genes or a new set of*
*highly expressed genes in testis.*

*Main text Line 301: Since genes with high expression in the testis are not uniformly distributed in the genome ^{8,13}, it*
*was possible that the reduced expression of the X and 4th chromosomes was due to the absence of genes highly*
*expressed in spermatocytes rather than a chromosome-wide reduction in expression due to a more global inactivation.*
*A way to avoid this potential confounding effect, is to explore the expression of widely expressed “housekeeping”*
*genes. We explored three data-driven methods to determine X and 4th chromosome expression of genes with*
*housekeeping functions. In the first two methods, we used low tissue-specificity genes based on Tau and Tissue*
*Specificity Score (TSPS) using our data ^{68,69}. The third method was a more granular low cell-type specificity metric*
*within in the scRNA-seq experiments (CTSP). Specifically, a set of widely expressed genes expressed in $\geq 33\%$ of all cells.*
*These methods reduced the expressed gene set numbers to varying degrees, with CTSP being the most stringent (Table*
*1). The Y chromosome was expressed in an exquisitely tissue-specific matter and has no widely expressed genes using*
*any metric. To determine if the functions of these three reduced gene sets are consistent with generic gene function,*
*we systematically analyzed Gene Ontology (GO) enrichment for all three subsets of genes (Fig. 4A; Table S5). There*
*are differences in function in the three gene sets. For example, in the Molecular ontology, enzymes were enriched in*
*Tau and CTSP gene sets, while regulators (which are less likely to be generic) were more enriched in the Tau gene set.*
*In the Biological ontology, all three sets were enriched for protein metabolism, consistent with “housekeeping”, but*
*the tissue-level Tau and TSPS gene sets were enriched for genes with development and female gamete functions,*
*which is not commonly thought to be generic. Housekeeping genes are often highly expressed. All the reduced gene*
*sets had higher median expression than all expressed genes, but elevated expression was most pronounced in the*
*CTSP gene set (Fig. 4B-E). Additionally, the CTSP gene set showed greater uniformity in expression levels across cell*
*types. Based on these results, we concluded that the CTSP gene set was the best subset for exploring expression of*
*“housekeeping” genes.*

*We then used the reduced gene sets to examine expression of the X and 4th chromosomes in all testis cell types.*
*Importantly, when we examined relative expression, all three reduced gene sets showed significantly reduced X/A*
*expression in germline cells (Fig. 4F-H). However, Tau and TSPS gene sets showed reduced X expression in all cell types*
*resulting in X/A ratios approaching 0.5 in both somatic and germline cells (Fig 4F,G). At face value, failed dosage*
*compensation might be expected to approach 0.5. These observations suggest a reason for the previous conclusion*
*that there is no dosage compensation in male germline ¹¹ following the analysis of widely expressed genes using*
*tissue-specificity scores. This conclusion is likely spurious, as testis somatic cells express the dosage compensation*
*genes (File S1) and the protein complex decorates the X in those somatic cells as occurs in X-chromosome dosage*
*compensation in other somatic cells ²³. In contrast, the CTSP gene set showed reduced X chromosome expression,*
*approaching 0.5, only in the late primary spermatocytes (Fig. 4H). Spermatogonia and somatic cells showed X/A*
*ratios approaching 1.0. Like the analysis of all expressed genes (Fig. 3), the parsimonious explanation is that*
*spermatogonia show dosage compensation and spermatocytes show inactivation or reduced X chromosome*
*compensation.*

*We similarly examined expression of the reduced gene sets for the 4th chromosome. Genes with low Tau were over-*
*represented on the 4th chromosome, especially in M1^o germ cells and C1 somatic cells, resulting in an exaggerated*

over-expression relative to the major autosomes across all cell types (Fig. 4H), while low TSPS and CTSP resulted in
 significantly lower relative expression of the 4th chromosomes only in spermatocytes (Fig. 4I,J). The magnitude of
 spermatocyte decrease was magnified when we used the CTSP gene set, but overall the 4/A ratios were near 1.0 (Fig.
 4J). The large sample size of cells resulted in tightly centered distributions, but note that the number of genes
 contributing the 4th chromosome measurements was small (Table 1). To briefly summarize, we observed a decrease in
 X and 4th chromosome expression with all genes (Fig. 3) and with reduced gene sets (Fig. 4), suggesting a
 chromosome-wide change in gene expression in spermatocytes, and not simply a reduced number of X-linked and 4-
 linked genes with male-biased expression.

 While there is the above new text associated with figure 4, some less processed data is also
 informative. Below is a plot of the chromosome element distribution of all expressed genes, low
 $\tau_{\alpha\omega}$, low TSPS, and low CTSP genes in each of the testis cell types. These are cell-level results
 (so all chromosome elements are significantly different due to sample size, thus no * shown), but
 you can clearly see the under-representation of X-linked genes in EVERY cell type using the
 tissue specificity metrics in the vertically stretched image below. The effect is shown in main
 paper in the ratiometric Figure 4 panels.

There are some interesting patterns here for future exploration, as the $\tau_{\alpha\omega}$ and TSPS gene sets
 might well inform sexual selection in non-germline cells. Indeed, we also saw non-germline
 effects in the reproductive tract (Figure 1). We raise this, but not expansively.

*Main text Line 124: The reproductive tract pattern of X chromosome expression is difficult to explain by absence of*
*germline X chromosome dosage compensation or meiotic sex chromosome inactivation, since there are no germ cells*
*in this tissue. By elimination, this suggests that sexual selection drives gene expression patterns of X-chromosome*
*expression in the reproductive tract.*

*Main text Line 328: However, Tau and TSPS gene sets showed reduced X expression in all cell types resulting in X/A*
*ratios approaching 0.5 in both somatic and germline cells (Fig 4F,G).*

7. Page 7, paragraph 3: “. There was a significant and progressive decrease ($P \leq 0.001$) in steady-
state...”. The legend typed as $P \leq 0.01$. They should be same.

Good catch. Possible confusion, due to multiple different significance cutoffs we used in the
original paper, caused us to reevaluate p-values in the manuscript. We have now settled on $p \leq$
0.01 throughout the manuscript. Every * is $p \leq 0.01$. Every “significant” statement in the main
text is at $p \leq 0.01$.

8. Page 19, the last paragraph needs to recompose: the Y chromosome issue above; the last
sentence “ ‘sex chromosome nature’ could be a conserved aspect ...” does not mean much.

The other reviewers also found this characterization of the 4th chromosome objectionable too.
The end of the paper has been completely rewritten to more expansively highlight the retention
of inactivation of the 4th and is more nuanced with respect to gene-by-gene versus chromosome-
wide violation of sex chromosome inactivation by the Y in the new model (figure 7) and
associated text.

*Main text line 448: We observed two violations of the prediction that unpaired chromosomes are silenced in primary*
*spermatocytes. Specifically, the 4th chromosome would be active, and the Y would be inactive in the simplest versions of*
*this model. However, the 4th has retained its X chromosome-like silencing despite having two copies and the Y is*
*maximally expressed in spermatocytes despite having a single copy. One way to achieve this would be the creation of a*
*repressed territory occupied by both the X and 4th chromosomes (Fig. 7C). The evolutionarily retained inactivation of*
*the 4th could be due to this localization, perhaps originally triggered by monosomy in ancestral species. It is also*
*possible that the non-recombining 4th chromosome is not recognized as having a homolog. The single Y is highly*
*diffuse and very little of it is in this repressed territory. However, allele-specific expression of the Y-linked rRNA genes*
*drive the activity of the nucleolus, so at least part of the Y is expressed while in a repressed territory. It is possible*
*that Y-linked genes, including the rDNA cluster, required for spermatogenesis escape inactivation as occurs for a*
*subset of X linked genes on inactive X chromosomes in mammals.*

Reviewer #3 (Remarks to the Author):

In this paper, Mahadevaraju et al performed single-cell RNA-seq of drosophila testes, focusing
their analysis on the sex chromosomes X and Y, as well as chromosome 4, an autosome derived
from an ancient X chromosome. The study of sex chromosomes is important for understanding
evolutionary processes specific to them and how these differ from the autosomes. Sex
chromosomes also demonstrate interesting regulatory mechanisms that can serve as a paradigm
for understanding gene regulation in general. As well as providing these insights, the field is
essential to understanding infertility and sex-biased diseases.

The authors defined testicular cell populations by comparing gene expressions with published

spatial expression data. They showed that chromosomes X and 4 have reduced gene expression
compared to chromosomes 2 and 3 in primary spermatocytes. Immunofluorescence analysis
suggested reduced activation of RNA Polymerase II being the cause of this repression. The paper
reports a new resource of single-cell transcriptome in drosophila larval testes. A number of
analyses needs to be refined to support authors' conclusions. Comments and suggestions for
experiments are listed below.

This is an excellent summary of our effort. Thank you for the thoughtful comments on
clarification and encouraging us to show more of the work.

Major comments:

1) general comment: The novelty and significance of the work need to be clarified more to
justify strong impact in the field. I appreciate that this work provides a useful resource for
drosophila testis biology, but scRNA-seq of drosophila testes was already done (Witt et al,
eLIFE, 2019) and the concept of X chromosome inactivation in male germ cells is not novel
(reviewed in Vibranovski, J Genomics, 2014).

We hope that the data speak for us and are confident that this manuscript will be widely cited.
Our contribution is a focused, high-resolution analysis, and a mechanism for a very specific and
important problem of sex chromosome expression. Briefly, for this paper, even though the
methodology and the problem are not new, our intellectual contribution is important and novel.
We have important new observations, a molecular model that explains them, and provide
insights that will help drive the field forward.

We did not cite Witt et al in the original submission, which was a major oversight on our part.
We were certainly aware of this work, as members of both teams discussed our data sets at the
Dallas Drosophila (March 2019) meeting for example. They have every right to be upset about
this (now corrected) oversight.

*Main text line 255: Our data (See Fig. S2 for UMAP projections for each Drosophila gene), along with a similar*
*dataset from adult testis ⁶⁴, should be an outstanding resource for those studying testis development and physiology.*

It is possible that the reviewer would like us to analyze the Witt et al data. While the raw data
from Witt et al. are up at the SRA, analyzed data, such as cell type calls, are not publicly
available (you would think eLife would insist on this), which means that someone interested in
Witt et al. cell type calls would have to repeat the analysis (which we have done, but do not show
here). We are not blaming Witt et al, as where scRNA-Seq analyzed data and metadata should go
is unresolved. Currently, data is in labs, at repositories (our data and metadata are up at GEO,
SRA, etc, See Table S6), and/or on sharing sites (we used
<https://doi.org/10.35092/yhjc.11950746> for images that would be difficult at GEO). Not having
Witt et al. analyzed data is problematic for us. Cell type calls are highly dependent on batches
and precise parameter settings, so our re-analysis of Witt et al was done exactly as in our paper
rather than theirs. We have some disagreements with them on some of the cell types, especially
the hub cell calls, which we do not find convincing, but hasten to point out that this is a feature
of scRNA-seq data, not any shortcoming in the Witt et al analysis, which focused on new genes.

However, we did see sex chromosome element expression patterns fully consistent with our data
using the Witt et al data, as redone by us. We are not interested in comparing datasets for this
work, and expect that the generating lab should be the one to publish any chromosome element
work. Indeed, our (admittedly rigorous) interpretation of the Toronto genomic data sharing
agreement (Toronto International Data Release Workshop Authors. Prepublication data sharing.
Nature. 2009 Sep 10;461(7261):168-70) precludes this analysis without making it collaborative.
Additionally, we are involved in the Fly Cell Atlas project, which will soon be the key resource.
We prefer to focus our attention moving forward rather than cross-validating.

2) page 4, fig 1: What is the significance of comparing gene expressions between testis and
ovary? To show the testis-specific inactivation of chromosomes X and 4, other tissues should be
included in this analysis.

Thank you for suggesting showing these additional experiments. As stated, we showed the
comparison of larval testis and larval ovary to demonstrate that the inactivation of X and 4 was
testis-specific, rather than leaving the possibility open that this occurs in gonads of both sexes.
Given the importance of the sex-specificity in the manuscript, this is an essential argument that
requires showing data. Adding other tissues, to show the complexity of X and 4th expression
relative to non-gonadal tissues, does have added value. We have brought the RNA-seq data set
size to 128 samples in the current manuscript summarized in the new figure 1. New text for this
figure is shown in the response to reviewer #1 and below (and of course in the paper). Briefly,
we now show data from whole adults, female and male heads, thorax (minus viscera), abdomen
(minus viscera, gonads, and reproductive tract), viscera, and reproductive tract from both the
*w1118* and *OreR* strains. There are sex chromosome biases in other tissues, but the gonads are
unique in showing significant differences in expression of both sex chromosomes and the former
sex chromosome 4.

*Main text line 106: Drosophila have X and Y sex chromosomes, two major autosome pairs and a pair of “dot” 4th*
*chromosomes (Fig. 1A). The Y and 4th chromosomes are gene poor, while the remaining chromosome arms are gene*
*rich (Fig. 1B). To examine sex-biased gene expression patterns, we focused on the distribution of male-biased gene*
*expression across chromosomes or chromosome arms (chromosome elements) for each tissue. We measured adult gene*
*expression (quadruplicates) in the whole body (Fig. 1C) as well as seven tissues (Fig. 1D-J): head, thorax (viscera*
*removed), abdomen (viscera and all reproductive organs removed), viscera (including digestive and excretory organs),*
*reproductive tract (gonads and genitalia removed), terminalia (including genitalia and analia), and gonads in females*
*and males from two strains. We found a significant deviation from random (we use $p \leq 0.01$ throughout this study) in 6*
*sample types, including the whole body, head, thorax, viscera, reproductive tract, and gonad (χ^2 test of independence).*
*To examine which chromosome elements contribute to this non-randomness, we performed a post hoc analysis (χ^2 test)*
*for each chromosome element (Table S1). Sex chromosomes and former sex chromosomes are the major contributors*
*to the non-randomness.*

*For X-chromosomes, we observed underrepresentation of male-biased gene expression in the whole body from either of*
*two wildtype strains (Fig. 1C), as previously reported ^s. In heads, we observed a slight enrichment in male-biased gene*
*expression in one strain (Fig. 1D). In contrast, we observed a reduction in male-biased gene expression in the*
*reproductive tract (Fig. 1H). The reproductive tract pattern of X chromosome expression is difficult to explain by*
*absence of germline X chromosome dosage compensation or meiotic sex chromosome inactivation, since there are no*
*germ cells in this tissue. By elimination, this suggests that sexual selection drives gene expression patterns of X-*
*chromosome expression in the reproductive tract. In the gonads, we observed an underrepresentation of male-biased*
*gene expression (Fig. 1J), as previously reported ^s.*

*Males with no Y chromosome are viable, but sterile and the Y chromosome is known to be expressed in spermatocytes*
*³⁰. However, the tissue-specific Y chromosome gene expression pattern is poorly described. We report that Y-*
*chromosome gene expression was detectable only in whole males and gonads (Fig 1C, J).*

*The 4th chromosome showed a decrease in male-biased gene expression in the whole body in one strain (Fig. 1C), an*
*increase in male-biased gene expression in the thorax in one strain (Fig. 1E), and most strikingly, a decrease in male-*
*biased expression in the gonads of both strains (Fig 1J). As a former X chromosome, 4th chromosome expression in the*
*gonads was especially interesting as it mirrored the X-chromosome underrepresentation of male-biased gene*
*expression. Additionally, and unlike the X chromosome, 4th chromosomes are present in two copies in males. Because*
*there are two copies of the 4th chromosome genes, under-representation of male-biased expression cannot be explained*
*by the absence dosage compensation. In summary, only gonads show sex-biased expression of the Drosophila X, Y, and*
*4th chromosomes.*

3) page 7: Discussion of comparison of X chromosome and autosomes:

- It is said that “Expression of the single X chromosome relative to the major autosomes
(chromosomes 2 and 3, each present in two copies) is not significantly different in
spermatogonia or any of the somatic cell types using either all expressed genes or widely
expressed genes (Fig 3A, B)”. In this sentence it is not clear what the statistical comparison is
relative to. In the figure caption this seems to be described as relative to the average of the
somatic cells.

We stated in the original text that we made multiple comparisons, but we have reemphasized this
now. One comparison is spatial and involves comparing among cell types in the dataset and
the other is temporal within the germline lineage. Both are important. We have done every
pairwise comparison and show the statistics in Table S5.

*Main text line 259: We looked at the dynamics of sex chromosome gene expression in germ cells in addition to all the*
*other cell types from the single cell dataset (Fig. 3, Table S5) Expression of the single X chromosome relative to the*
*major autosomes (chromosomes 2 and 3, each present in two copies) was not significantly different in spermatogonia,*
*early primary spermatocytes or any of the somatic cell types (Fig. 3A).*

*Main text line 266: Nevertheless spermatogonia showed similar levels of X chromosome expression relative to*
*autosomes, providing new evidence for non-canonical dosage compensation of the X chromosome in pre-meiotic germ*
*cells. There was a significant decrease in expression of the X chromosome in early, middle and late primary*
*spermatocytes (M1° and L1°) relative to either spermatogonia or somatic cells.*

*Main text line 270: This decrease in X expression approached 2-fold which could be due to either a loss of dosage*
*compensation in germ cells as they mature into primary spermatocytes, or to the gain of meiotic X-chromosome*
*inactivation during the transition from mitotic spermatogonia to meiotic spermatocytes.*

*Main text line 277: Expression of 4th chromosome genes parallels what was seen for the X (Fig. 3B). The expression*
*ratio of the two 4th chromosomes relative to the two sets of major autosomes hovered near 1. There was a significant*
*decrease in relative 4th chromosome expression in middle and late primary spermatocytes (M1°, and L1°) compared to*
*expression in either spermatogonia or somatic cells.*

*Supplemental text line 39: Table S5. Gene sets and chromosome elements. Consists of six parts: a readme, distribution*
*of genes among Chromosome elements, pairwise comparisons of global expression between different cell types using*
*different gene sets (all expressed, CTSP, Tau, and TSPS), GO analysis of gene sets, X/A, 4/A, and Y/A ratio significance*
*testing, pairwise significance testing for all chromosome elements by gene set. Supporting data for Table 1, Fig 1B, Fig*
*2E, and Fig 4.*

Are they saying that gonads are not significantly different to the average of somatic cell types, and
that also each somatic cell type is not significantly different to the average of somatic cell types?
This should be explained better. Additionally, I’m not sure that the latter assertion is
informative.

Yes. We hope this is now clear in the text above. We agree that saying that there is no
difference in X chromosome expression among somatic cell types is expected, but we do want to

show this. This becomes more important now, given that we are showing that using the sets of
genes widely expressed among tissues (Tαv and TSPS) show exactly this type of unexpected
differences in X/A ratios in somatic cells, due to global reductions in expression of these gene
sets in every cell type in the testis (Figure 4). Tαv and TSPS have additional problems as outline
in the text.

*Main text line 326: We then used the reduced gene sets to examine expression of the X and 4th chromosomes in all testis*
*cell types. Importantly, when we examined relative expression, all three reduced gene sets showed significantly reduced*
*X/A expression in germline cells (Fig. 4F-H). However, Tav and TSPS gene sets showed reduced X expression in all*
*cell types resulting in X/A ratios approaching 0.5 in both somatic and germline cells (Fig 4F,G). At face value, failed*
*dosage compensation might be expected to approach 0.5. These observations suggest a reason for the previous*
*conclusion that there is no dosage compensation in male germline \perp following the analysis of widely expressed genes*
*using tissue-specificity scores. This conclusion is likely spurious, as testis somatic cells express the dosage*
*compensation genes (File S1) and the protein complex decorates the X in those somatic cells as occurs in X-*
*chromosome dosage compensation in other somatic cells ²³. In contrast, the CTSP gene set showed reduced X*
*chromosome expression, approaching 0.5, only in the late primary spermatocytes (Fig. 4H). Spermatogonia and*
*somatic cells showed X/A ratios approaching 1.0. Like the analysis of all expressed genes (Fig. 3), the parsimonious*
*explanation is that spermatogonia show dosage compensation and spermatocytes show inactivation or reduced X*
*chromosome compensation.*

- It is said that “There was a significant and progressive decrease ($P \leq 0.001$) in steady-state
expression of the X chromosome in early, middle and late primary spermatocytes (E1°, M1°, and
L1°).” Does the statistical test actually show there is a progressive decrease, or is the test just
showing that each of the spermatocytes is independently significantly less than the average of the
somatic cells? If the former, the testing need to be explained better, if the latter, the assertion
needs to be clarified.

Yes, as stated in the revised text above, we were referring to the temporal decrease within the
germline, in addition to expression relative to the somatic cells. You may note that the E1° are
no longer marked as being significantly reduced. This is due to our decision, outlined earlier, to
use only $p \leq 0.01$ throughout the manuscript. E1° was significant at $p \leq 0.05$.

- It is said that “Expression of 4th chromosome genes paralleled what was seen for the X. There
was a significant and progressive decrease ($P < 0.001$) in steady-state expression levels in M1°
and L1° (Fig. 3C, D) compared to expression in spermatogonia.” It seems that the tests for the X
were performed relative to the somatic cells, not spermatogonia – so if the testing is relative to a
different cell type for chr4, there is not a ‘parallel’ present. However, there is confusion as to
what the testing is relative to: the figure caption suggests also that for the 4th chromosome the
tests were done relative to the somatic cells – so the mention of significant decrease relative to
the spermatogonia here is a confusing one which should be clarified.

We have made this clarification as for the X in the preceding comment and response. The 4th
chromosome expression in germ cells was treated exactly the same.

*Main text line 277: Expression of 4th chromosome genes parallels what was seen for the X (Fig. 3B). The expression*
*ratio of the two 4th chromosomes relative to the two sets of major autosomes hovered near 1. There was a significant*
*decrease in relative 4th chromosome expression in middle and late primary spermatocytes (M1°, and L1°) compared to*
*expression in either spermatogonia or somatic cells. Since we can rule out failed dosage compensation as a cause of 4th*
*chromosome decreased expression, there must be a gain of inactivation during the developmental transition from*

*mitotic spermatogonia to meiotic spermatocytes. This X chromosome like behavior may reflect the evolutionary history*
*of the 4th chromosome; specifically, that the 4th retained X-inactivation after reacquiring autosomal status.*

4) page 8: It is said that “We observed poor expression of the Y in somatic cells, and increased
expression in E1^o, M1^o, and L1^o primary spermatocytes. This is likely to occur from expression
of a few highly transcriptionally active Y-linked genes originally identified by the cytologically
visible Y-chromosome loops present at these stages (64, 65).” If the authors have the data
available, can they not test this likely scenario and confirm or deny it?

Y-linked gene expression has been directly probed cytologically by others and we do have the
gene-level expression data for the Y, so we can in fact confirm that this is what is happening.

*Main text line 287: We observed poor expression of the Y in somatic cells and spermatogonia, and increased*
*expression in primary spermatocytes (E1^o, M1^o, and L1^o), concomitant with decreased expression of the X and 4th*
*chromosomes. This occurs from expression of a 42 transcriptionally active Y-linked genes, consistent with the diffuse*
*chromatin and Y-loops originally identified by cytology of primary spermatocytes 66,67.*

5) page 8: It is said that “The decrease in sex chromosome expression in M1^o and L1^o did not
reflect an overall decrease in total gene expression compared to somatic lineages”. However, it
has just been shown that Y chromosome expression does not decrease.

Thank you for catching this. We did just show that Y chromosome expression increased. We
have used “X and 4th” rather than “sex”.

*Main text line 287: We observed poor expression of the Y in somatic cells and spermatogonia, and increased*
*expression in primary spermatocytes (E1^o, M1^o, and L1^o), concomitant with decreased expression of the X and 4th*
*chromosomes.*

Previous assertions for chrX and chr4 do not show a decrease in sex chromosome expression,
they show a decrease in expression relative to the autosomes, which is a different measure.

We show X/A and 4/A expression in most of the figures and do not directly show the A
expression, but the ratio metric results are due to the X and 4th is accurate. We have made liberal
use of the term “relative” as shown in the example below. The figures clearly indicate
ratiometric measures.

*Supplemental text line 39: Table S5. Gene sets and chromosome elements. Consists of six parts: a readme, distribution*
*of genes among Chromosome elements, pairwise comparisons of global expression between different cell types using*
*different gene sets (all expressed, CTSP, Tau, and TSPS), GO analysis of gene sets, X/A, 4/A, and Y/A ratio significance*
*testing, pairwise significance testing for all chromosome elements by gene set. Supporting data for Table 1, Fig 1B, Fig*
2E, and Fig 4.

*Main text line 259: Expression of the single X chromosome relative to the major autosomes (chromosomes 2 and 3,*
*each present in two copies) was not significantly different in spermatogonia, early primary spermatocytes or any of the*
*somatic cell types (Fig. 3A). The somatic cell X chromosome expression relative to autosomes hovered near 1.0 despite*
*the 2-fold dose difference, a pattern consistent with the known canonical X chromosome dosage compensation*
*mechanism in somatic cells, expected to increase expression from the single X₂. This dosage compensation mechanism*
*does not exist in germ cells. Nevertheless spermatogonia showed similar levels of X chromosome expression relative to*
*autosomes, providing new evidence for non-canonical dosage compensation of the X chromosome in pre-meiotic germ*
cells.

6) page 8, fig 4C-F: Overlap with DAPI-dense region doesn't necessarily correlate to inactivation
as 2L-euc-positive region is also DAPI-dense (fig 4E).

While that particular image is probably overexposed to show the territory, we have not
systematically quantified DAPI levels in territories. We have therefore deleted all statements
about DAPI density.

Does any heterochromatin marker protein specifically localise on the chr X/4 territory?

There is literature showing that a whole host of proteins localize in or near the nucleolus,
including some interesting players such as spermatocyte-specific transcriptional machinery and
Pc. There will likely be an interesting story here. We cite some of those papers (especially the
Rob White lab) when they directly related to our work on Pol-II. We will want to repeat
qualitative results in order have quantification and correlation with expression, Phospho-CTD,
etc. While there are a lot of interesting reasons to do more extensive experiments on what is
present on these chromosomes, especially as it might relate to Pol-II promoter clearance and
elongation, those experiments are beyond the scope of this paper.

7) page 9, fig 4G-H: Volume should be normalised by chromosome length covered by each
probe. Chromosomes 3 and 4 should be included in these data.

Thank you for pointing out our failure to be clear. The first step in this normalization was
actually in the probe selection, as we stated in the original submission. Further normalization
therefore does not really change in a way that is detectable in the figure, but we also did length
normalization in the original submission. We have now stated this explicitly.

*Main text line 380: We probed similar sized euchromatic regions of the X chromosome (22.3 Mb) and the left arm of*
*the 2nd chromosome (2L, 22.7 Mb) with oligopaints (Fig. 5E).*

*Main text line 384: We found a that probe length corrected X chromosome volume was reduced relative to chromosome*
*2L (Fig. 5H).*

We have also added a new figure panel that adds normalization for copy number in addition to
normalization by length.

*Main text line 386: However, when we corrected the data to account for 2L copy number (divided by two), the X had*
*significantly greater volume than 2L (Fig. 5I) which was inconsistent with inactivity resulting from compaction.*

Chromosomes 3 and 4 should be included in these data.

We do not feel that adding chromosome 2R, 3L, 3R would add significantly to our knowledge,
only to our work. In contrast, we really would like to look at the 4th post-pandemic., although
volume and sphericity measurements would probably not be our highest priority when thinking
about the structure of the 4th.

Minor comments:

8) general comment: Please insert line numbers to help reviewers to refer points.

Done -- our apologies for not doing this from the beginning.

9) general comment: Please use colour-blind friendly colouring in figures.

We have improved the color coding by proofing using both protanopia and deuteranopia filters (these are easy to use in illustrator, which we will now use routinely). Many of the bars and boxplots were clearly problematic. These have been corrected. The original figure 1 was particularly difficult, so we simply rendered the new and improved figure 1 in gray scale. In figure 2, we changed the color coding for T (from purple to light yellow) and P (brown to bright yellow). These color-coding changes propagate through the box plots in later figures. Given the large number of colors used in the clustering of cell types, this is still not perfect, but it is much improved. Thank you.

10) general comment: Throughout the paper the ‘Seq’ in ‘scRNA-seq’ is capitalised. This is not how it is found in the literature and should be changed to ‘seq’ to match.

“RNA-Seq” often is capitalized and it seems strange to have “RNA-Seq” and “scRNA-seq” in the same paper, so we now have lower case “seq” everywhere. We have no real preference here and defer to the copy editor if the manuscript is accepted.

11) general comment: Throughout the paper there is a mixture of ‘Fig N’ and ‘Fig. N’. These should be changed to be consistent throughout.

We have used “Fig. N” throughout. We are unsure about which format the journal prefers and will defer to the copy editor if the manuscript is accepted

12) page 3: It is not clear why they are using L3 larvae instead of adults and what benefit/questions this brings. Previous work has done scRNA-seq on adult testis (eg. Witt et al, eLIFE, 2019). Maybe something specific about Drosophila biology makes this important? This should be made clear for readers. Should also perhaps be mentioned in the abstract at least?

We used larvae to avoid microfluidic and filter fouling due to sperm, which are very long cells (this turned out not to be a problem) and to enrich for spermatogonia and primary spermatocytes relative to the vast numbers of secondary spermatocytes and sperm which were of little interest for this work, as we stated in the original. We expanded this explanation slightly.

Main text line 148: We decided to use single cell RNA sequencing (scRNA-seq) ³¹ for a higher resolution picture. We did not want to use read depth to sequence transcriptionally inactive secondary spermatocytes and sperm, so we selected Drosophila third instar larval (L3) testis for our experiments. They contain abundant germ cells, including the critical transition from mitotic spermatogonia to meiotic primary spermatocytes ^{32,33}.

We added “larvae” to the abstract, but we are unclear if that is what was requested. The adult versus L3 choice is peripheral methodology, not the main result or intellectual contribution, so we really don’t want to say more.

*Main text line 39: Using single cell RNA-Seq on larvae, we demonstrate that the single X and pair of 4th chromosomes*
*are specifically inactivated in primary spermatocytes, based on measuring all genes or a new set of highly expressed*
*genes in testis.*

13) page 4: “expression of male-specific Y chromosome was highly testis-biased” is an unusual
thing to conclude since Y chromosome is only present in testis, there can be no ‘bias’ relative to
ovaries in the standard sense of the word.

Agreed. We still use this terminology but in the context of the other male tissues (Figure 1).

*Main text line 130: Males with no Y chromosome are viable, but sterile and the Y chromosome is known to be*
*expressed in spermatocytes ³⁰. However, the tissue-specific Y chromosome gene expression pattern is poorly described.*
*We report that Y-chromosome gene expression was detectable only in whole males and gonads (Fig 1C, J).*

14) page 4: "We identified 18,965 single cells across three biological replicates (Spearman $\rho \geq$
0.93 , $P < 0.001$; Table S1)". In this context I do not think it is clear what the Spearman's rank
refers to. It sounds like it is somehow related to the number of cells when placed after the current
sentence, while examining Table S1 shows it is related to gene expression ranks.

Thank you. Clarified as requested.

*Main text line 182: We identified 18,965 single cells across three biological replicates based on the intersection of*
*calls from cell ranger count ⁴⁰ and DropletUtils emptyDrops ⁴¹ (Table S2). Potential cell doublets were detected using*
*scrublet ⁴² and removed. Based on preliminary cluster analysis using the 2,000 most variably expressed genes, we set*
*the perplexity threshold in Seurat ⁴³ to 0.3. This yielded ten clusters, each potentially representing a distinct cell type or*
*state with each of three biological replicates contributing to the clusters (Spearman expression rank correlation ≥ 0.93 ,*
*$p < 0.01$, Fig. 2B).*

15) page 5: replacing “RNA-seq” with “bulk RNA-seq” would make this explanation clearer,
especially when it is mentioned right after scRNA-seq.

Yes. This is a good idea. We have gone through the entire manuscript and ensured that the
distinction between bulk and single cell profiles are clear.

16) page 7: “widely expressed genes” are defined as genes expressed in $> 33\%$ of all cells in the
single cell data. Is expression of these genes biased to specific cell types?

The short answer is no. We have added a new figure 4 that shows the characteristics of the
widely expressed genes, which we now call low Cell Type SPecificity genes (CTSP) and
contrast to two other gene subsets designed to investigate generically expressed genes.

[revised manuscript text omitted]

17) page 7: “Drosophila tissue”  “Drosophila tissues”

Done. We rechecked the entire document for grammatical number category.

18) page 7: “dosage compensation” is used to refer to X upregulation in a number of places
throughout the manuscript. The term “dosage compensation” generally is used to refer to both X
upregulation and X chromosome inactivation mechanisms, and so care should be taken not to use
the generic term in describing just one of the mechanisms it encompasses, particularly when
talking about both of them in the same paragraph.

Agreed. In flies, dosage compensation generally refers to upregulation of the X, but this is not
always the case. We have gone through the manuscript and checked every occurrence of dosage
compensation to make sure it is clear if we observe/expect/cite up-regulation or inactivation.

19) page 7: Please define "steady state expression" of chrX. Is "steady state expression" a term
used commonly to refer to "expression relative to autosomes"? If so, this is fine. If not, then it
could be easily misunderstood that their data is showing the transcription from the X absolutely
decreases, when in fact it is the ratio of transcriptional activity between X and autosomes that is
shown to be decreasing.

We used steady-state to indicate that we were measuring transcripts, not transcription. Gene
expression analysis is a misused term that implies that transcription is being measured, it is not.
Our best data on actual transcription versus steady-state transcript levels is the activated pol-II
data. Obviously, we were not clear. We have just dropped the term "steady-state" throughout
and tried to write more plainly in each of those locations.

20) page 8, second paragraph: How often does chr 4 localise in the region including chr X?
Please add the conclusion of this section.

Thank you. These data were, and are, in the supplement, but not in the main text, where they are
quite useful. This is now shown in a new panel (Figure 5F) and new text.

*Main text line 370: We observed that the X was universally near the nucleolus (median distance 0.2 μm) and the 4th
was nearly as close (median distance 0.7 μm), well within the same prominent territory (Fig. 5D,F). Since the 4th
occupies the same territory as the X, these chromosomes could be regulated independently, or coordinately, due to
territory-level regulation.*

21) page 8: the sentence "Spermatocyte chromosomes are represented (Fig.4)." does not make
sense in isolation, I think this sentence has been accidentally inserted.

Thank you. Deleted.

22) page 8: I think "X chromatin heterochromatic satellite sequences" should read "X
chromosome heterochromatic satellite sequences".

Thank you. Written as suggested

*Main text line 362: In situ hybridization reveals X chromosome heterochromatic satellite sequences near the
prominent spermatocyte nucleolus, where Ribosomal DNA repeats are located, and ribosome biogenesis occurs (Fig.
5B,C,F) 24.*

23) page 10: "This suggests that sex chromosome, not copy number, determines activity in
primary spermatocytes.". This sentence does not make sense in this form. I think the sentiment is
"This suggests that some property intrinsic to sex chromosomes modulates their expression in a
way independent of copy number"?

We have completely rewritten the close of the manuscript, which now features a model figure for
clarity.

*Main text line 430: Mechanistically, the reduced expression of the X and 4th chromosomes in spermatocytes correlates
with the failure to activate RNA Pol-II. The Y chromosome is concomitantly active. This beg the question, why? This
expression pattern could be due to the simple absence of genes expressed in spermatocytes on the X and 4th
chromosomes and the presence of genes that must be expressed from the Y chromosome (Fig 7A). If there are few*

*genes expressed, there will be little active Pol-II ipso facto. However, expression of “housekeeping” genes suggests a*
*chromosome-wide decrease in X and 4th expression. A prediction of a pure gene content model is that genes newly*
*arriving on the X with would be expressed, as evolutionary modification of regulation takes time. In fact, the autosomal*
*ocnus gene is precisely expressed in spermatocytes, but shows extremely reduced reporter activity when inserted onto*
*the X²⁸. This is consistent with a model where the X is a generally unfavorable environment for spermatocytes gene*
*expression, due to either chromosome- or territory-level repression (Fig. 7B,C). This is reminiscent of meiotic sex*
*chromosome in mammals, where X chromosome expression is high in spermatogonia, followed by X inactivation*
*associated with a distinct organelle like XY body¹⁵. The inactivation of both the X and Y chromosomes in mammals may*
*be a special case of a more general inactivation of unpaired chromosome regions in a genomic defense model¹⁶. Lack*
*of homology could signal intruding transposable elements seeking to hijack the germline for vertical transmission to*
*the next generation. Active recognition and silencing would be useful to the host organism. We observed two violations*
*of the prediction that unpaired chromosomes are silenced in primary spermatocytes. Specifically, the 4th chromosome*
*would be active, and the Y would be inactive in the simplest versions of this model. However, the 4th has retained its X*
*chromosome-like silencing despite having two copies and the Y is maximally expressed in spermatocytes despite having*
*a single copy. One way to achieve this would be the creation of a repressed territory occupied by both the X and 4th*
*chromosomes (Fig. 7C). The evolutionarily retained inactivation of the 4th could be due to this localization, perhaps*
*originally triggered by monosomy in ancestral species. It is also possible that the non-recombining 4th chromosome ⁴ is*
*not recognized as having a homolog. The single Y is highly diffuse and very little of it is in this repressed territory.*
*However, allele-specific expression of the Y-linked rRNA genes drive the activity of the nucleolus²⁵, so at least part of*
*the Y is expressed while in a repressed territory. It is possible that Y-linked genes, including the rDNA cluster, required*
*for spermatogenesis escape inactivation as occurs for a subset of X linked genes on inactive X chromosomes in*
*mammals⁸². Interestingly, X to 2nd or 3rd chromosome translocations result in breakpoint-independent dominant male*
*sterility, whereas X to 4th do not⁷⁴. Spreading repression or activation along a chromosome element, or relocation of*
*parts of elements to novel territories might result in such a phenotype. Experiments to test these models will help us*
*understand the evolution of sex chromosome expression in flies, and probably many other species.*

24) page 10: the sentence “Where X-like chromosomes are inactivated and the Y-like
chromosomes are highly expressed.” does not make sense in isolation, I think this sentence has
been accidentally inserted.

Thank you. Deleted.

25) page 10: It is unclear what "sex chromosome nature" means.

We agree. See reviewer #3, comment #23 above.

26) figure 1:
- X and Y chromosomes should be next to each other on the axis.

Done

- Y axis label should explain better what the measure is (I think ‘average gene expression’, not
just ‘expression’).

We agree that this was insufficient. However, this was/is not a gene level measurement. We
have looked at expression at the “location” level of arm or chromosome in cells. We have made
this clearer in the legend and relabeled the Y axis. We have in general followed a descriptive
text axis label and then (usually parenthetically) the units of measure. So: Gene Expr. Per Cell
(gene density normalized). Clarity here is obviously critical. Thank you very much for pointing
this out.

- X axis title of ‘chromosome arm’ is unsuitable since X and Y are not arms. Change to
something like ‘scaffold’ or ‘location’.

We agree, the X, Y, and 4th are chromosomes, not arms. The right arm of the X is negligible,
and we have not even attempted to distinguish the Y chromosome arms. Drosophila convention
often refers to arms as chromosomes (e.g. Chromosome 2L), which is also inaccurate. Scaffolds
would be problematic due to gaps. Location seems vague. Drosophila convention uses elements
to refer to these combinations of Chromosomes and Arms, which is what we have adopted in the
revision.

*Main text line 108: To examine sex-biased gene expression patterns, we focused on the distribution of male-biased*
*gene expression across chromosomes or chromosome arms (chromosome elements) for each tissue.*

27) figure 2:

- D-I: bottom right panel is difficult to read: Having it as a line graph does not make sense as the
data is not a series. Bar graphs should be used instead. X axis labelling being only on the last
panel makes it hard to read for other panels, if barchart was used with bars colour-coded to
match the cell types as in panel A/B, this may be clearer.

We agree. Thank you for this suggestion. It is easier to read as a color coded barchart, which we
have adopted.

- E: line graph suggests highest expression in Gonias and E1°, but IF image seems to show higher
expression in M1°/L1° (based on the cartoon in panel A)?

We agree. However, this is not an error. The IF images are protein-traps. There is a great deal
of translational control in the male germline, which we have clarified. Protein expression
following the appearance of the mRNA was scored as overlap.

*Main text line 225: In addition, we show that ADD domain-containing protein 1 (Add1), which encodes a*
*heterochromatin associated protein that interacts with Heterochromatin Protein 1 (HP1) to maintain heterochromatin*
*52,53, has enriched expression in early primary spermatocytes in the scRNA-seq data. The Add1 protein accumulated*
*throughout spermatocyte stages, consistent with translational control and/or protein stability over several days of germ*
*cell development (Fig. 2G).*

*Supplemental text line 257: We compared gene expression patterns between scRNA-Seq and manually curated images*
*(43 genes from the literature; 31 genes from this study; Table S4). We developed an overlap score (0-4) between*
*scRNA-Seq expression and mRNA or protein expression in curated images. A score of 4 indicates a gene showed cell*
*type biased expression (scRNA-Seq) in the exact same cell types as mRNA or protein expression (curated images).*
*Since protein expression may lag transcription, we also gave a 4 when protein expression was later in the specific cell*
*lineage. A score of 3 indicates gene was highly expressed, but not cell type biased, in the exact same cell types as*
*mRNA or protein expression. A score of 2 indicates cell type biased gene expression in the same cell lineage as mRNA*
*or protein expression. Finally, a score of 1 indicates high gene expression in the same cell lineage as mRNA or protein*
*expression.*

28) figure S1: there are no scale bars on panels A and B.
The scale bars have been added.

REVIEWERS' COMMENTS

Reviewer #1 (Remarks to the Author):

In this new draft, the authors have significantly revised the text and figures to make the message clearer and to provide more direct evidence for their conclusions. The revisions fully address my previous concern about the cause of the differences in X chromosome gene expression in germ cells. In addition, the revisions help to highlight their interesting observations in support of dosage compensation in pre-meiotic germ cells, despite the lack of dosage compensation machinery in these cells, and chromosome silencing at later stages of germ cell differentiation. The revisions also address my question about whether the scRNAseq data align with expectations from bulk RNAseq and clear up the sections I found confusing. Therefore, I now fully support publication of the manuscript.

Reviewer #2 (Remarks to the Author):

I am satisfactory with the revision, which corresponded convincingly to all my criticisms, main ones or minors or suggestions. Especially, I like their clarification of the Y at the various levels to address the questions I asked. I recommend to publish as it is. I am glad the fields of Drosophila genetics and evolution in general have one more solid and important observation now, which makes good sense of previous works and theories in the studies of the related scientific issues.

Reviewer #3 (Remarks to the Author):

The revised manuscript has additional data and analyses and many of the points raised by the reviewers have been dealt with. However, the lack of the depth in the analyses on the mechanism of chromosome X/4 silencing in testes limits the significance of the study (see my comment in point a-1).

a) comments on authors' response (numbers refer to the comment number of reviewer 3):

1) The authors agree that their scRNA-seq approach of drosophila testis and the concept of X silencing is not novel. They claim that they "have important new observations, a molecular model that explains them". Although the silencing of chromosome 4 is a new finding, insights into the molecular mechanism of the silencing derive only from Pol II phosphorylation data in Fig 6. The authors declined to do additional analyses raised in the point 6. Since the main novelty of this paper is about the mechanism of testis-specific silencing of chromosomes X/4, it should have been analysed more deeply.

2) The authors added new data analysing sex-bias in gene expression in tissues (Fig 1 C-K), but this data would not be relevant to support authors' conclusion. To show testis-specific low transcription from chromosomes X/4, they should have simply compared gene expression levels between chromosomes in each tissue in male and females, not sex bias.

3) The authors have made the figure caption for Fig 3 clear now, but the main text is still not clear. Lines 259-262 needs to be changed to say that the significant difference is against the (average of?) somatic cell types. To say that there is a difference but not to explain what it is different to makes no sense. It is a simple fix: e.g. "Expression of the single X chromosome was not significantly different *to the average of somatic cells* in spermatogonia, early primary spermatocytes....". In fact, this whole sentence is redundant since I think the following sentences (line 262-276) explain the same information in a more understandable way. I would probably recommend omitting the first sentence (beginning "Expression of the single X chromosome....")

and just retaining the following ones.

4) It is not clear where the data corresponding to the 42 transcriptionally active genes that the authors describe is (line 289-291). I think it is in supplemental table S2 but looking at this it is not clear where the number 42 comes from. Please reference the data in the text and clarify (in supplemental is fine) where the number comes from.

6) See comments to the point 1

b) Additional minor comments on the rewrite:

1) Fig 1: adding element labels to the bottom of each graph, while cluttering it up a bit, would make this figure much easier to understand

2) Line 307: It is not clearly defined what ' $T_{\alpha\upsilon}$ ' (T, alpha, upsilon) means. Ref #69 introduces 'Tau', a tissue-specificity metric which can also be abbreviated to the Greek letter tau (τ), but nowhere before have I seen this written as ' $T_{\alpha\upsilon}$ ' so I am unsure what it means. I might guess that this is a misunderstanding of the name of the metric, and if so it should be changed throughout the text, tables and figures. If it is a new measure then this should be communicated more clearly in the methods/text.

3) I noted during my checking of the tau methods that the referencing is muddled in the supplementary material. For example Ref #68 is cited when discussing tau in the 'scRNA-seq Downstream Analysis' section of the methods in the supplement – in fact the paper which I think they should reference (and do correctly in main paper) is #69 in the main paper or #62 in the supplementary. I have not checked other citations but these should all be checked carefully.

4) Fig 4: the figure caption does not match with the figure - I don't know what any of the lower panels refer to. Also the subpanels should be labelled on the figure with the specificity measure throughout to make it clearer.

5) Line 442-443: "This is reminiscent of meiotic sex chromosome in mammals..." – I think this should read "This is reminiscent of meiotic sex chromosome **inactivation** in mammals".

Please accept our sincere thanks for the reviewer time and effort put into improving our manuscript. The full REVIEWERS' COMMENTS (black) and our responses (blue) are below. We have used track changes in the main text and supplement and underlined new text in this response.

Be well.
For the authors,

Brian Oliver

Reviewer #1 (Remarks to the Author):

In this new draft, the authors have significantly revised the text and figures to make the message clearer and to provide more direct evidence for their conclusions. The revisions fully address my previous concern about the cause of the differences in X chromosome gene expression in germ cells. In addition, the revisions help to highlight their interesting observations in support of dosage compensation in pre-meiotic germ cells, despite the lack of dosage compensation machinery in these cells, and chromosome silencing at later stages of germ cell differentiation. The revisions also address my question about whether the scRNAseq data align with expectations from bulk RNAseq and clear up the sections I found confusing. Therefore, I now fully support publication of the manuscript.

Thank you for your help. Your careful reading and thoughtful suggestions were critical to improving the manuscript.

Reviewer #2 (Remarks to the Author):

I am satisfactory with the revision, which corresponded convincingly to all my criticisms, main ones or minors or suggestions. Especially, I like their clarification of the Y at the various levels to address the questions I asked. I recommend to publish as it is. I am glad the fields of Drosophila genetics and evolution in general have one more solid and important observation now, which makes good sense of previous works and theories in the studies of the related scientific issues.

Thank you for your time and effort. We appreciate your enthusiasm for the significance in terms of resolving some theoretical aspects and extending the field of sex chromosome biology.

Reviewer #3 (Remarks to the Author):

The revised manuscript has additional data and analyses and many of the points raised by the reviewers have been dealt with. However, the lack of the depth in the analyses on the mechanism of chromosome X/4 silencing in testes limits the significance of the study (see my comment in point a-1).

Thank you for your meticulous comments on our manuscript. We have made many of the changes suggested, which have contributed to improved readability.

a) comments on authors' response (numbers refer to the comment number of reviewer 3):

1) The authors agree that their scRNA-seq approach of drosophila testis and the concept of X silencing is not novel. They claim that they "have important new observations, a molecular model that explains them". Although the silencing of chromosome 4 is a new finding, insights into the molecular mechanism of the silencing derive only from Pol II phosphorylation data in Fig 6. The authors declined to do additional analyses raised in the point 6. Since the main novelty of this paper is about the mechanism of testis-specific silencing of chromosomes X/4, it should have been analysed more deeply.

We respectfully disagree with the proposition that the Pol-II phosphorylation is a trivial advance. Our opinion, and that of reviewers 1 and 2, is that the work represents a significant advance.

We agree that the suggested new experiments are interesting, but our rationale for pursuing them later are solid and we stand by this position. First, the paper is already a full read and story. We envision the proposed experiments as the start of the next phase of this line of research. Furthermore, as noted by the journal, the limitations on wet-bench activity due to Covid-19 are real, and the various institutes of the authors have all been affected.

2) The authors added new data analysing sex-bias in gene expression in tissues (Fig 1 C-K), but this data would not be relevant to support authors' conclusion. To show testis-specific low transcription from chromosomes X/4, they should

have simply compared gene expression levels between chromosomes in each tissue in male and females, not sex bias.

The reviewer's suggestion is a useful way of plotting the data and we have swapped out sex-biased expression for expression levels for the chromosome arms in figure 1. As suggested, this is a bit easier on the reader. The new figure is below. The main text and figure legend have been modified to reflect this change (these excerpts are shown below). The general message is unchanged: only gonads show sex-biased expression of the *Drosophila* X, Y, and 4th chromosomes.

Overall gene expression of the sex chromosomes varied by tissue. For X-chromosomes, we observed under-expression of genes in the whole body and gonads of males from either of two wildtype strains (Fig. 1C), as previously reported⁸. We also observed reduced X-chromosome expression in heads, thorax, reproductive tract, and terminalia (Fig. 1D-E,H). The non-gonadal patterns of X chromosome expression are difficult to explain by absence of germline X chromosome dosage compensation or meiotic sex chromosome inactivation, since there are no germ cells in those tissue. By elimination, this suggests that sexual selection drives gene expression patterns of X-chromosome expression in many somatic cell types. Under-representation in gonads could be explained in full or part by absence of dosage compensation or meiotic sex chromosome inactivation.

The 4th chromosome showed a decrease in gene expression in male whole bodies, reproductive tracts, and gonads (Fig. 1C, H, J). As a former X chromosome, 4th chromosome expression in the gonads was especially interesting as it mirrored the X-chromosome underrepresentation of male-biased gene expression. Additionally, and unlike the X chromosome, 4th chromosomes are present in two copies in males. Because there are two copies of the 4th chromosome genes, under-representation of expression cannot be explained by the absence of dosage compensation. These data again suggest that there has been sexual selection of sex chromosomes, but only gonads show sex-biased expression of the Drosophila X, Y, and 4th chromosomes.

Figure 1. Bulk RNA-Seq of seven adult tissues and L3 larval gonads

(A) Illustration of a Wild-type male karyotype cartoon depicting the size of the chromosomes and arms (chromosome elements) and the distribution of heterochromatin (black) and euchromatin (gray). (B) Haploid annotated gene content of chromosome elements (including non-coding genes). (C-K) For each tissue type we summed the transcripts per million reads (TPM) of each gene on a chromosome element and divided by the number of genes expressed on that arm. Male (black) and female (open) gene expression is shown. Adult tissues: (C) Whole body, (D) Head, (E) Thorax (viscera removed), (F) Abdomen (viscera, reproductive organs removed), (G) Viscera (digestive tract and malphigian tubules), (H) Reproductive tract (gonads and genitalia removed), (I) Terminalia (genitalia and analia), and (J) Gonad. For each tissue, we used two “wild-type” strains, *w¹¹¹⁸* and *Oregon-R (Ore-R)*, which are stacked in each panel. Late third instar larval gonads (K) are from *w¹¹¹⁸*. Significance at $p \leq 0.01$ is shown (*). Where the chromosomal expression showed a sex difference in both strains, the chromosome element is bold and in a larger font.

3) The authors have made the figure caption for Fig 3 clear now, but the main text is still not clear. Lines 259-262 needs to be changed to say that the significant difference is against the (average of?) somatic cell types. To say that there is a difference but not to explain what it is different to makes no sense. It is a simple fix: e.g. “Expression of the single X chromosome was not significantly different *to the average of somatic cells* in spermatogonia, early primary spermatocytes....”. In fact, this whole sentence is redundant since I think

the following sentences (line 262-276) explain the same information in a more understandable way. I would probably recommend omitting the first sentence (beginning “Expression of the single X chromosome....”) and just retaining the following ones.

We agree. That sentence has been deleted and the paragraph is now:

“We looked at the dynamics of sex chromosome gene expression in germ cells in addition to all the other cell types from the single cell dataset (Fig. 3, Dataset S5). The somatic cell X chromosome expression relative to autosomes hovered near 1.0 despite the 2-fold dose difference, a pattern consistent with the known canonical X chromosome dosage compensation mechanism in somatic cells, expected to increase expression from the single X²²..”

4) It is not clear where the data corresponding to the 42 transcriptionally active genes that the authors describe is (line 289-291). I think it is in supplemental table S2 but looking at this it is not clear where the number 42 comes from. Please reference the data in the text and clarify (in supplemental is fine) where the number comes from.

Thank you. We now specifically refer to the data in supplement and table as suggested (see below). The data are visible by sorting the table by chromosome element.

“This occurs from expression of a 42 transcriptionally active Y-linked genes (Supplemental text, Dataset S2), consistent with the diffuse chromatin and Y-loops originally identified by cytology of primary spermatocytes^{66,67}.”

6) See comments to the point 1

Addressed above

b) Additional minor comments on the rewrite:

1) Fig 1: adding element labels to the bottom of each graph, while cluttering it up a bit, would make this figure much easier to understand

We agree that the balance between clutter and clarity in labeling complex figures is critical. We have increased the labeling on this particular figure as suggested.

2) Line 307: It is not clearly defined what ‘ $T\alpha\upsilon$ ’ (T, alpha, upsilon) means. Ref #69 introduces ‘Tau’, a tissue-specificity metric which can also be abbreviated to the Greek letter tau (τ), but nowhere before have I seen this written as ‘ $T\alpha\upsilon$ ’ so I

am unsure what it means. I might guess that this is a misunderstanding of the name of the metric, and if so it should be changed throughout the text, tables and figures. If it is a new measure then this should be communicated more clearly in the methods/text.

Thank you for helping us clarify this. Tau is not a new metric and we fully reference in the text:

Line 308 “In the first two methods, we used low tissue-specificity genes based on Tau and Tissue Specificity Score (TSPS) using our data ^{68,69}.”

If the problem is capitalization, the metric has been capitalized (or not) in the literature. For example:

“We also analysed robustness of Tau by comparing correlation calculated on all 27 tissues and on all the subsets of 5–26 tissues (Supplementary Figures S12 and S13).”
<https://doi.org/10.1093/bib/bbw008>

Tau is a Greek letter and we used Greek font for Greek letters. We are happy to follow convention and journal formatting preferences as advised by the copy editor. We have added this as a query on the author checklist form.

3) I noted during my checking of the tau methods that the referencing is muddled in the supplementary material. For example Ref #68 is cited when discussing tau in the ‘scRNA-seq Downstream Analysis’ section of the methods in the supplement – in fact the paper which I think they should reference (and do correctly in main paper) is #69 in the main paper or #62 in the supplementary. I have not checked other citations but these should all be checked carefully.

Thank you for your careful observations. It is our understanding that the citation numbers will be collated into the main text references. We agree that having separate supplement and main text references is confusing and we probably should have combined them in the original draft. We will follow the journal guidelines and carefully check all citations in the proofs.

4) Fig 4: the figure caption does not match with the figure - I don’t know what any of the lower panels refer to. Also the subpanels should be labelled on the figure with the specificity measure throughout to make it clearer.

Thank you for catching this! We added a panel to this figure without changing the legend. We also added labels to Fig 4F-K, as suggested. The addition follows:

Line 1207 (a) Ribbons showing p-values for Gene Ontology enrichments (blue scale) in three gene sets representing widely expressed genes.

5) Line 442-443: “This is reminiscent of meiotic sex chromosome in mammals...”
– I think this should read “This is reminiscent of meiotic sex chromosome *inactivation* in mammals”.

Thank you. Corrected:

Line 443. This is reminiscent of meiotic sex chromosome inactivation in mammals, where X chromosome...